# Old Carbon, New Insights: Thermal Reactivity and Bioavailability of Saltmarsh Soils

Alex Houston[1], Mark H Garnett[2], and William E N Austin[1,3]

1. Department of Geography and Sustainable Development, University of St Andrews, St Andrews, KY16 9AL, United Kingdom
2. NEIF Radiocarbon Laboratory, Scottish Universities Environmental Research Centre, East Kilbride, G75 0QF, United Kingdom
3. Scottish Association of Marine Science, Oban, PA37 1QA, United Kingdom

*Correspondence to*: Alex Houston (ah383@st-andrews.ac.uk)

## Abstract

Saltmarshes are globally important coastal wetlands which can help to mitigate the impacts of climate change. They accumulate organic carbon from both modern and aged sources through in-situ biological production and the capture of ex-situ sources which are deposited during tidal inundation. Previous studies have found that long-term organic carbon storage in saltmarsh soils is driven by the net contribution from the older fraction, implying that the inputs of young organic carbon derived from in situ production are recycled at a faster rate.

Using ramped oxidation, we assessed the composition ($^{14}$C and $^{13}$C) of saltmarsh soil carbon pools defined by their thermal reactivity. By relating $^{14}$C measurements of the soil carbon pools to $CO_2$ respired in aerobic incubations of the same soils, we provide the first empirical evidence linking the thermal reactivity of saltmarsh soil organic carbon with its bioavailability for remineralization.

We found that old ($^{14}$C-depleted) carbon dominates the thermally recalcitrant organic carbon pools, whereas the thermally labile carbon is composed of younger organic carbon sources. In most cases, the $^{14}$C content of the most thermally labile carbon pool was closest to the previously reported $^{14}$C content of the $CO_2$ evolved from aerobic incubations of the same soils, implying that the bioavailability of saltmarsh soil organic carbon to remineralisation in oxic conditions is closely related to its thermal lability.

Our results highlight the importance of saltmarshes as stores of both old, thermally recalcitrant organic carbon, as well as younger, thermally labile organic carbon that is vulnerable to decomposition under oxic conditions. Management interventions (e.g. rewetting by tidal inundation)

to limit the exposure of saltmarsh soils to elevated oxygen availability may help to protect and conserve these stores of thermally labile organic carbon and hence limit $CO_2$ emissions. We also present the first evidence to support the inclusion of thermally labile allochthonous OC stored in saltmarsh soils in additionality assessments, with relevance to international carbon crediting projects and National GHG Inventories.

**Graphical Abstract**

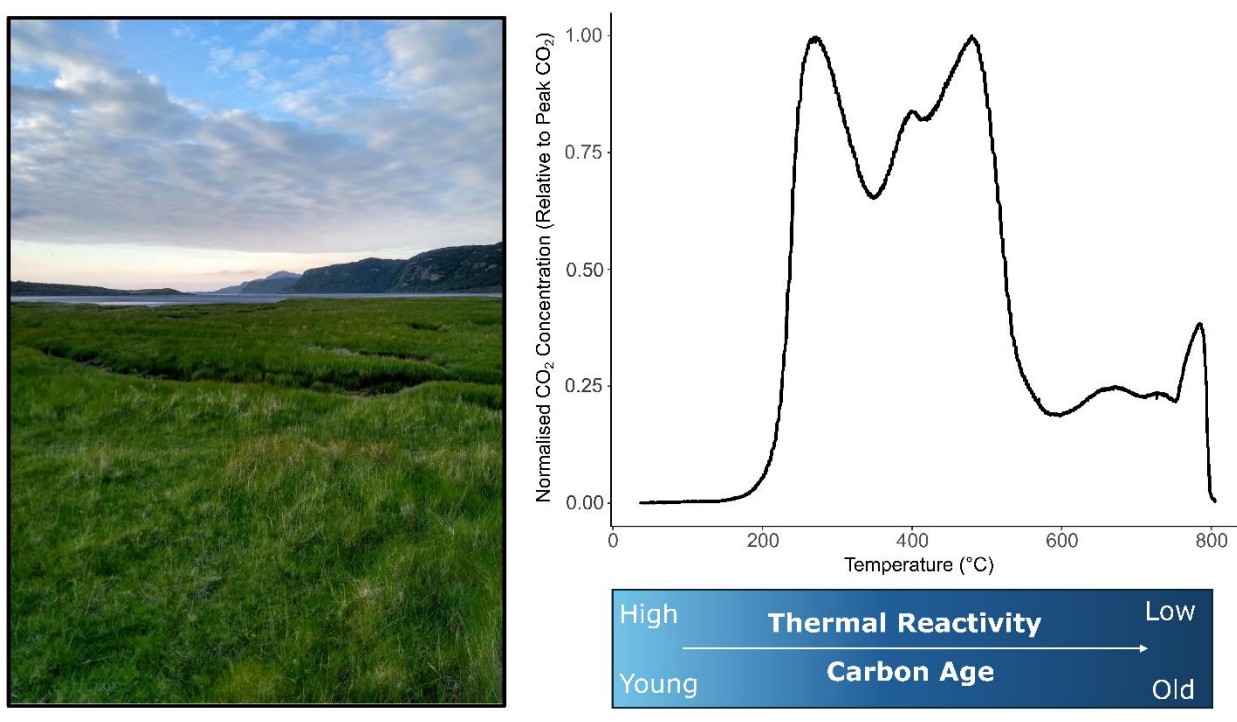

**1. Introduction**

Saltmarshes accumulate organic carbon (OC) of variable age and reactivity into their soils. A portion of this OC is stored for millennia, providing a climate regulation service, and some is returned to the atmosphere or laterally exported (Komada et al., 2022; Macreadie et al., 2021). Saltmarshes also accumulate and produce inorganic carbon (IC) but the climate regulation service of this is currently under debate and unclear (Granse et al., 2024; Van Dam et al., 2021).

To understand the role of saltmarsh soils in carbon cycling and their potential for climate mitigation through targeted management interventions, much research has focussed on determining the autochthonous (in-situ) and allochthonous (ex-situ, trapped during tidal inundation from terrestrial and marine sources) contributions to saltmarsh soils, with the accumulation of autochthonous OC as a direct sequestration of carbon from the atmosphere, reducing the amount of atmospheric greenhouse gases (GHGs) (Macreadie et al., 2019; Saintilan et al., 2013; Van de Broek et al., 2018). The accumulation of allochthonous OC, originally sequestered outside the saltmarsh area, does not directly reduce atmospheric GHGs, but can represent a source of avoided emissions if it remains stored in the saltmarsh soil for longer than in an alternative depositional environment (Howard et al., 2023). Evidence to determine whether this is the case or not, and under what scenarios, has proven challenging to obtain (Geraldi et al., 2019; Houston et al., 2024a). OC pools with distinct biological turnover times may instead provide greater insights into the soil carbon residence time and therefore the climate mitigation achieved through targeted management interventions to retain that carbon (Sanderman and Grandy, 2020).

Ramped oxidation (RO) and ramped pyrolysis oxidation (RPO) have been used to estimate the thermal reactivity and biological turnover time of soil and sediment OC (Hemingway et al., 2017b; Plante et al., 2011; Rosenheim et al., 2008). RO and RPO involve measuring the quantity of $CO_2$ evolved as a sample is increasingly heated at a constant rate in an atmosphere containing oxygen (e.g., Plante et al., 2011; Stoner et al., 2023), or other gases, typically Helium (RPO: e.g., Hemingway et al., 2017a; Rosenheim et al., 2008). The temperature at which $CO_2$ is thermally-evolved is related to the activation energy required to thermally decompose C (Hemingway et al., 2017b), which is also an estimate of the energy required for biological degradation of OC (Peltre et al., 2013; Plante et al., 2013). $CO_2$ evolved at low temperatures is deemed to be from soil OC pools with a greater thermal lability than $CO_2$ evolved at higher temperatures (Peltre et al., 2013; Rosenheim et al., 2008). OC thermal reactivity pools can be examined by collecting the evolved $CO_2$ from set temperature ranges with distinct thermal reactivities and measuring the $^{14}C$ (age)

and $^{13}C$ content (Rosenheim et al., 2008), which can then be related to the activation energy
required to thermally decompose those C sources (Hemingway et al., 2017b).
The $^{14}C$ content of the thermal reactivity pools provides insight into the turnover time of each pool,
with past research showing that the oldest soil organic matter (OM) (most depleted $^{14}C$ content)
tends to dominate the most thermally recalcitrant fractions (Bao et al., 2019b; Plante et al., 2013;
Stoner et al., 2023). Similar results have been found for saltmarsh soils (Luk et al., 2021). Young
OC, which can be autochthonous or allochthonous (Van de Broek et al., 2018), has been found
to turnover at a faster rate than old OC in saltmarsh soils (Komada et al., 2022; Van de Broek et
al., 2018), implying that young OC may tend to be more thermally labile than old OC for saltmarsh
soils.
The $^{13}C$ content of the thermal reactivity pools can also provide insight as to whether the source
of OC has an influence on turnover time. Previous work has found that the $^{13}C$ content of evolved
$CO_2$ tends to be more enriched at higher temperatures due to greater contributions from $^{13}C$-
enriched, degraded/microbially derived OC (Luk et al., 2021; Sanderman and Grandy, 2020;
Stoner et al., 2023). Similarly, comparisons of the isotopic composition of thermally-defined OC
pools to their chemical properties have found that thermally labile OC is derived from mostly
lipids and polysaccharides, whereas OC with a higher thermal recalcitrance is derived from a
greater proportion of phenolic and aromatic compounds (Sanderman and Grandy, 2020). The
thermal reactivity of soil and sediment OC is also influenced by the formation of organo-mineral
complexes, which can physically and chemically stabilise OC (Bianchi et al., 2024; Hemingway
et al., 2019). Mineral-associations can increase the energy required for decomposition and have
been found to increase thermal recalcitrance and to slow turnover times of soil and sediment OC
(Hemingway et al., 2019; Stoner et al., 2023).
Crucially, the biological availability (bioavailability) of OC for decomposition, and hence its
biological turnover time, depends on the prevailing environmental conditions as well as thermal
reactivity (Hemingway et al., 2017b; Schmidt et al., 2011). For example, increased hydrodynamic
energy can destabilise organo-mineral complexes and increase the bioavailability of previously
stable OC (Spivak et al., 2019). Similarly, increased oxygen availability can decrease the energy
requirement for microbes to decompose molecularly recalcitrant OC, causing it to be
remineralised at a faster rate (Noyce et al., 2023).
Houston et al. (2024b) found that young OC stored in saltmarsh soils was preferentially respired
as carbon dioxide ($CO_2$) during aerobic incubation experiments, but that a portion of the respired
$CO_2$ was produced from an aged ($^{14}C$-depleted), allochthonous source.  It is possible that this
$CO_2$ could have been respired from thermally labile as well as thermally recalcitrant soil OC
sources because the increased oxygen availability of the incubations potentially facilitated the
degradation of OC which was previously stable in the low-oxygen environment of typical
saltmarsh soils (Noyce et al., 2023).
The isotopic composition of RO thermal reactivity fractions can be compared to the isotopic
composition of the $CO_2$ that is evolved biologically during incubations of equivalent samples to
determine whether or not the age of the most biologically- and thermally-reactive OC pools
match. Here, we present the first measurements of the $^{13}C$ and $^{14}C$ content of $CO_2$ derived from
saltmarsh soils using RO, and the first comparison of these to the $^{14}C$ content of biologically
evolved $CO_2$ from the same soils (Houston et al., 2024b). We hypothesised that the thermally
labile C pools would be composed of younger C than the thermally recalcitrant pools, and that
the $CO_2$ evolved from saltmarsh soils exposed to oxic conditions (Houston et al., 2024b) are from
a predominantly thermally labile OC pool.
**2. Methods**
**2.1. Field site and sample collection**
Three saltmarsh soil cores (T1-3) were retrieved ca. 30 m apart from the lower marsh zone from
Skinflats (SK), an estuarine saltmarsh in Scotland (56° 3'34.04"N, 3°43'59.16"W), as detailed in
Houston et al. (2024b). Field methods and laboratory sub-sampling procedures are described in
detail in Houston et al. (2024b). Briefly, the cores were split into 1 cm thick slices as follows: core
T1 (0-1 cm, 5-6 cm, and 18-19 cm); T2 (0-1 cm, 5-6 cm, and 15-16 cm), and T3 (0-1 cm, 5-6 cm,
and 19-20 cm) (with the deepest sample from each core being the deepest retrieved sample from
the 20 cm length of the corer. On the occasions when a full core was not retrieved, the deepest
retrieved soil was used). Each slice was subsequently divided to provide sample material for the
RO procedure, and for aerobic laboratory incubations from which the biologically evolved $CO_2$
was collected for $^{13}C$ and $^{14}C$ analysis (Houston et al., 2024b).
**2.2. Ramped oxidation**
The RO sub-samples were individually dried to constant mass before milling to a fine powder to
homogenise and limit potential shielding effects from aggregates. Unlike most RO and RPO
studies (e.g., Hemingway et al., 2017b), we did not remove carbonates from our samples. Acid
treatment, which is required to remove carbonates from samples has been demonstrated to
result in losses from the labile OC fraction (Bao et al., 2019a). A loss of labile OC for our samples
could seriously impact the interpretations in our study, and our ability to compare the $^{14}C$ content
of the $CO_2$ respired from bulk (untreated) soils in the incubation experiments (Houston et al.,
2024b) to the [14]C content of the RO thermal fractions.
The samples were sent to the NEIF Radiocarbon Laboratory for RO, which is described in Garnett
et al. (2023). The RO procedure involved two stages, a first combustion to determine the
relationship between the rate of $CO_2$ evolution and temperature (thermogram), and a second
combustion where evolved sample gases were collected across defined temperature ranges, for
subsequent isotope analysis. For the first combustion, ca. 200 mg of dried and homogenized
sample material was weighed into a quartz vial which was inset into a quartz combustion tube,
which was subsequently placed into a furnace set initially to room temperature. The furnace was
progressively heated at a constant rate of 5°C per minute to 800°C in a stream of high purity
oxygen (N5.5, BOC, UK). Heating caused combustion of the sample and the evolution of gas
which was passed into a second quartz combustion tube containing platinised wool in a furnace
set to a constant temperature of 950°C. The platinised wool acted as a catalyst to ensure
complete combustion of the evolved gases. Upon exiting the secondary combustion chamber
the sample passed through a glass tube containing magnesium perchlorate desiccant to remove
moisture and subsequently the $CO_2$ concentration of the gas was measured using a non-
dispersive infrared $CO_2$ sensor (SprintIR®-WF-5, Gas Sensing Solutions, UK). The sample was
then passed out of the sensor unit and vented to the atmosphere.
The measured $CO_2$ concentration (normalised for sample mass) was plotted against temperature
to produce thermograms which were used to identify temperature ranges, which defined C
thermal reactivity pools for this study: 150-325 °C, 325-425 °C, 425-500 °C, 500-650 °C, and 650-
800 °C.
For each sample, the required mass of material to evolve sufficient $CO_2$ (> 3 mL) for [14]C
measurement was calculated based on the thermogram. A new sub-set from the original dried
and homogenised sample was then re-run following the RO procedure outlined above, but
instead of venting to atmosphere, after its measurement the evolved $CO_2$ was collected into foil
gas bags based on the defined temperature ranges. $CO_2$ was collected for [13]C analysis from 650-
800 °C, but sufficient $CO_2$ was evolved for [14]C analysis from this thermal fraction for only one
sample (T1 0.5 cm, Table A1) and we do not consider this fraction further because it is likely
dominated by carbonates and not relevant to the purpose of this study.
The foil gas bags (5 L Spout Pouch, https://www.pouchshop.co.uk/) used for sample collection
were sealed with one-hole rubber bungs into which a 0.6 cm diameter x 5 cm length stainless
steel tube was inserted. Isoversinic tubing (Saint Gobain, France) was fitted over the stainless
steel to connect it to a quick coupling (Colder Products Company, USA), which allowed
connection to the RO kit.
Prior to the RO $CO_2$ collection, all equipment was cleaned using a standardised procedure
(Garnett et al., 2023). All glassware was combusted at 900°C for a minimum of two hours, and all
couplings and connectors were washed in carbon-free detergent (Decon) and rinsed in Milli-Q
water. The foil gas bags were cleaned by repeatedly (3 times) filling with ca. 1 L high purity nitrogen
gas (Research Grade 99.9995% purity, BOC, UK) and evacuating with an air pump, over a period
of at least 24 hours (to aid out-gassing of $CO_2$). The final evacuation, immediately before
connecting to the RO rig, involved pumping out the bags with an SBA-5 $CO_2$ analyser (PPsystems,
USA) to ensure that the bags did not contain significant contamination. Before commencing a
sample combustion, the entire RO rig was checked for leaks and other potential sources of
contamination by measuring the $CO_2$ concentration in the oxygen carrier gas exiting the kit, using
the SBA-5 $CO_2$ analyser.
Within 3 days of combusting a sample, the evolved gas in each foil bag was connected to a
vacuum rig for cryogenic recovery of pure sample $CO_2$ by passing it through slush (–78°C; dry ice
and industrial methylated spirits) and then liquid nitrogen (–196°C) traps, under high vacuum (ca.
$3 \times 10^{-3}$ millibars). The sample $CO_2$ was then split into three aliquots: One for $\delta^{13}C$ analysis using
isotope ratio mass spectrometry (IRMS; Delta V, Thermo-Fisher, Germany), one for graphitisation
and subsequent AMS $^{14}C$ analysis, and one for an archive back-up. The graphitised AMS samples
were measured for $^{14}C$ content at the SUERC AMS Laboratory (see Ascough et al., 2024). The $^{13}C$
content ($\delta^{13}C$-VPDB) was used to normalise the $^{14}C$ results to a $\delta^{13}C$ of -25 ‰ to correct for
isotopic fractionation. Following convention, $^{14}C$ results are presented as %Modern (fraction
modern x 100) and conventional radiocarbon ages (years BP, where 0 BP = AD 1950 and age = -
8033 x Ln (%Modern/100)).
**2.3. Data Analysis**
Continuous activation energy distributions were modelled from thermograms using the
'*rampedpyrox*' package in Python V3.8 (Hemingway, 2016; Hemingway et al., 2017b). The
*rampedpyrox* model calculates mean activation energies ($\mu E$) and the standard deviation of
activation energy ($\sigma E$), which is a measure of the heterogeneity of bond strength, for each
temperature fraction which $CO_2$ was collected from. Mean $\mu E$, $\sigma E$ and activation energy
distribution (p (o,E)) are also calculated for each sample using the *rampedpyrox* model. We do
not use the *rampedpyrox* model for calculation of isotope values as it applies a blank correction
to $^{14}C$ (Hemingway et al., 2017a, b) which is not relevant to the analytical set-up for this study
(Garnett et al., 2023), and the $^{13}$C values generated varied significantly from our IRMS measured
values (Table A2). Further data analysis and visualisation of thermograms and isotopic data was
undertaken using RStudio V4.2.2 (R Core Team, 2022).

## 3. Results

### 3.1. Radiocarbon

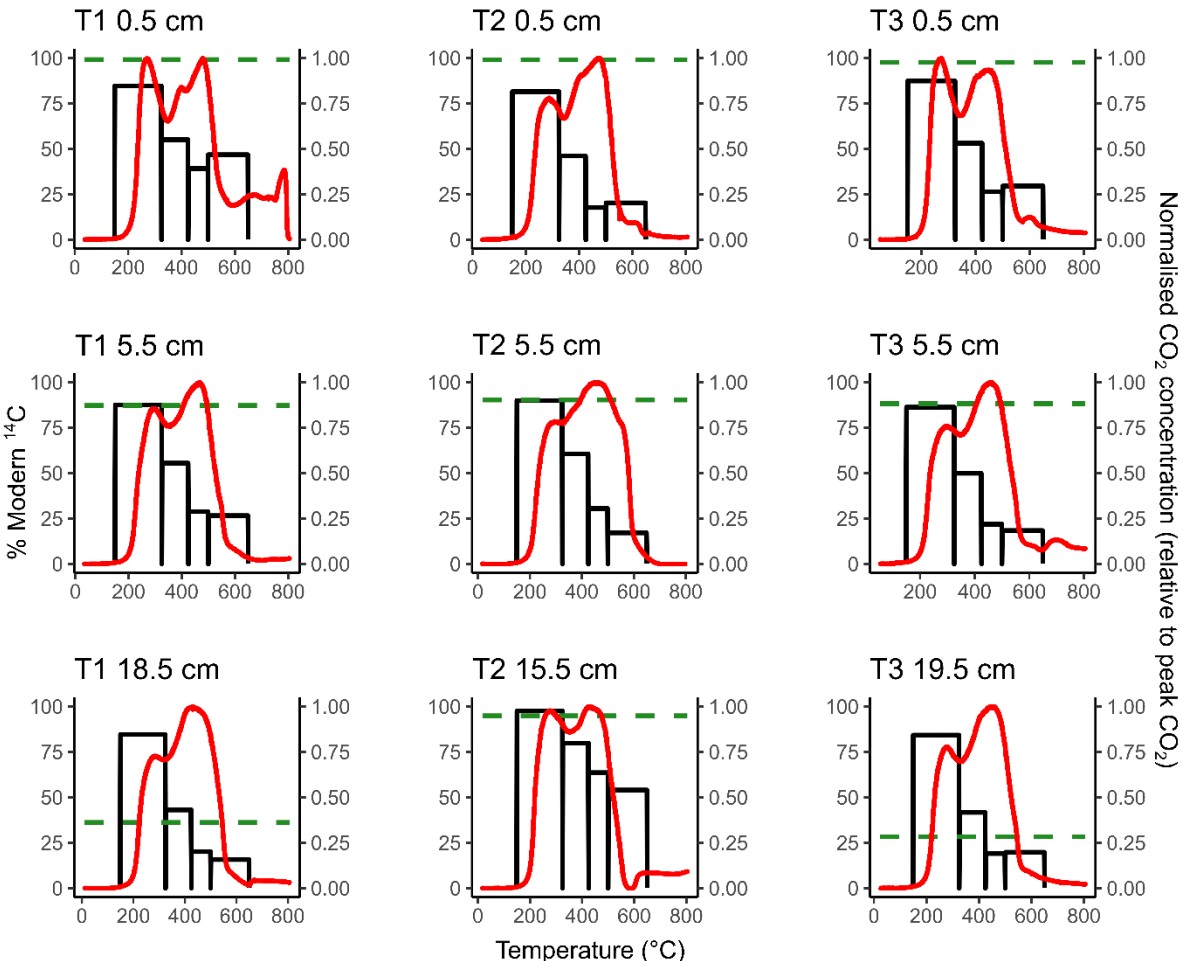

*Figure 1. Thermograms (red lines, right-hand y-axis) overlaying the $^{14}$C content of ramped oxidation fractions (black bars, left-hand y-axis) for each sample. The horizontal green dashed lines represent the $^{14}$C content of the CO$_2$ respired from the aerobic incubation experiments of Houston et al. (2024b).*

The $^{14}$C content of the RO fractions (Fig. 1, Table 1) were statistically similar between the 0.5 cm,
5.5 cm, and deepest sample (T1 18.5 cm, T2 15.5 cm, T3 19.5 cm) depth increments for each of
the temperature fractions (Kruskal-Wallis; p = 0.83, 0.38, 0.66, 0.99, for 150-325°C, 325- 425°C,
425-500°C, 500-650°C, respectively). There were, however, clear differences in $^{14}$C contents
between the temperature fractions, with ranges of 81.50-97.54 % Modern for 150-325 °C, 41.67-
79.80 % Modern for 325-425 °C, 17.67-63.56 % Modern for 425-500 °C, and 15.69-53.96 %
Modern for 500-650 °C (Fig. 1, Table 1).
*Table 1. Radiocarbon concentration (% Modern) of RO temperature fractions and the $CO_2$*
*produced in soil incubation experiments in Houston et al. (2024b). Errors are reported to one*
*standard deviation from the mean. A sole $^{14}C$ measurement for T1 0.5 cm 650-800 °C is reported*
*in Table A1.*

| | % Modern $^{14}C$ | | | | |
|---|---|---|---|---|---|
| | 150-325°C | 325-425°C | 425-500°C | 500-650°C | Incubation $CO_2$ (Houston et al., 2024b) |
| **T1 0.5 cm** | 84.62 ± 0.44 | 55.02 ± 0.29 | 39.18 ± 0.21 | 46.75 ± 0.26 | 99.15 ± 0.45 |
| **T1 5.5 cm** | 87.51 ± 0.43 | 55.43 ± 0.28 | 28.76 ± 0.17 | 26.56 ± 0.16 | 87.18 ± 0.38 |
| **T1 18.5 cm** | 84.56 ± 0.44 | 43.06 ± 0.23 | 20.07 ± 0.13 | 15.70 ± 0.12 | 36.13 ± 0.36 |
| **T2 0.5 cm** | 81.50 ± 0.43 | 46.04 ± 0.24 | 17.67 ± 0.13 | 20.26 ± 0.14 | 98.97 ± 0.43 |
| **T2 5.5 cm** | 89.95 ± 0.42 | 60.55 ± 0.30 | 30.54 ± 0.17 | 17.11 ± 0.12 | 90.26 ± 0.40 |
| **T2 15.5 cm** | 97.53 ± 0.50 | 79.80 ± 0.41 | 63.56 ± 0.31 | 53.96 ± 0.27 | 94.86 ± 0.44 |
| **T3 0.5 cm** | 87.37 ± 0.45 | 53.09 ± 0.28 | 26.37 ± 0.15 | 29.55 ± 0.17 | 97.56 ± 0.43 |
| **T3 5.5 cm** | 86.23 ± 0.42 | 49.86 ± 0.25 | 21.87 ± 0.14 | 18.36 ± 0.12 | 88.22 ± 0.41 |
| **T3 19.5 cm** | 84.23 ± 0.41 | 41.67 ± 0.22 | 19.04 ± 0.13 | 19.76 ± 0.14 | 28.25 ± 0.37 |


**3.2. $\delta^{13}C$**
There were no significant differences in the $^{13}C$ content of the ramped oxidation fractions (Fig. 2,
Table 2) between the depth increments (Kruskal-Wallis; p = 0.66, 0.63, 0.63, 0.44, 0.17, for 150-
325 °C, 325-400 °C, 425-500 °C, 500-650 °C, 650-800 °C respectively). $^{13}C$ contents followed the
opposite trend to $^{14}C$ contents with temperature, with ranges of -28.0 to -24.7 ‰ for 150-325 °C,
-26.6 to -22.3 ‰ for 325-425 °C, -25.4 to -20.2 ‰ for 425-500 °C, -24.4 to -13.9 ‰ for 500-650 °C,
and  -21.1 to -4.0 ‰ for 650-800 °C (Fig. 2, Table 2).

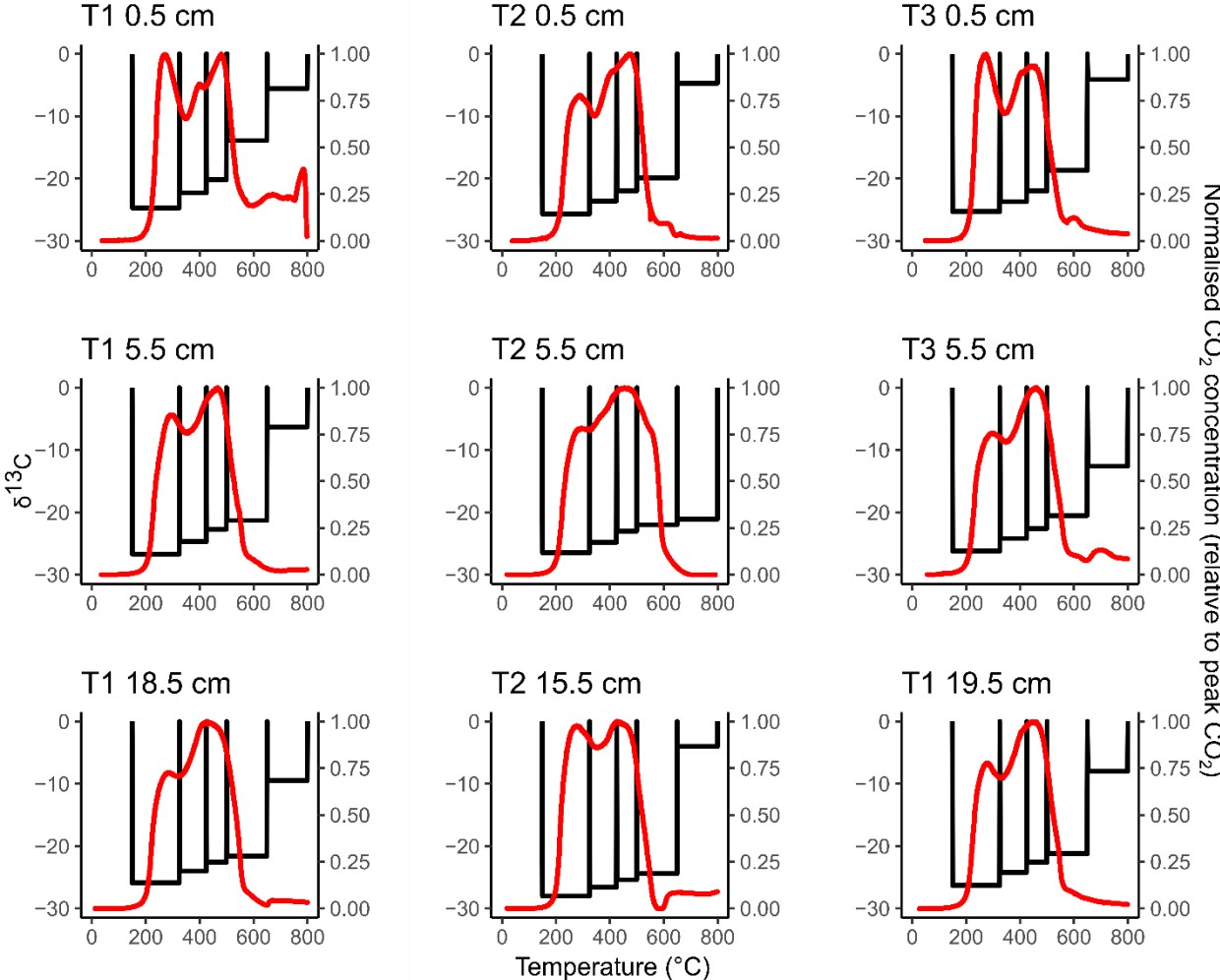


Figure 2. Thermograms (red lines, right-hand y-axis) overlaying the $^{13}C$ content of the RO temperature fractions (black bars, left-hand y-axis) for each sample. Unlike Fig. 1, we did not attempt to relate the $^{13}C$-RO to the $^{13}C$ content of the $CO_2$ respired in the incubation experiments, due to the potential for microbial fractionation during the incubation experiments.

199

Table 2. $\delta^{13}$C-VPDB‰ signature of the RO temperature fractions and the incubation experiments in Houston et al. (2024b). Errors are reported to one standard deviation from the mean.

| | $\delta^{13}$C-VPDB‰ | | | | | |
|---|---|---|---|---|---|---|
| | 150-325°C | 325-425°C | 425-500°C | 500-650°C | 650-800°C | Incubations (Houston et al., 2024b) |
| T1 0.5 cm | -24.7 ± 0.1 | -22.3 ± 0.1 | -20.2 ± 0.1 | -13.9 ± 0.1 | -5.6 ± 0.1 | -23.3 ± 0.1 |
| T1 5.5 cm | -26.7 ± 0.1 | -24.7 ± 0.1 | -22.7 ± 0.1 | -21.3 ± 0.1 | -6.3 ± 0.1 | -23.6 ± 0.1 |
| T1 18.5 cm | -25.9 ± 0.1 | -24.0 ± 0.1 | -22.6 ± 0.1 | -21.6 ± 0.1 | -9.5 ± 0.1 | -6.1 ± 0.1 |
| T2 0.5 cm | -25.7 ± 0.1 | -23.6 ± 0.1 | -22.0 ± 0.1 | -19.9 ± 0.1 | -4.7 ± 0.1 | -22.9 ± 0.1 |
| T2 5.5 cm | -26.5 ± 0.1 | -24.8 ± 0.1 | -23.0 ± 0.1 | -22.0 ± 0.1 | -21.1 ± 0.1 | -23.1 ± 0.1 |
| T2 15.5 cm | -28.0 ± 0.1 | -26.6 ± 0.1 | -25.4 ± 0.1 | -24.4 ± 0.1 | -4.0 ± 0.1 | -20.2 ± 0.1 |
| T3 0.5 cm | -25.3 ± 0.1 | -23.7 ± 0.1 | -22.0 ± 0.1 | -18.7 ± 0.1 | -4.1 ± 0.1 | -20.6 ± 0.1 |
| T3 5.5 cm | -26.2 ± 0.1 | -24.2 ± 0.1 | -22.6 ± 0.1 | -20.5 ± 0.1 | -12.6 ± 0.1 | -23.4 ± 0.1 |
| T3 19.5 cm | -26.3 ± 0.1 | -24.2 ± 0.1 | -22.6 ± 0.1 | -21.2 ± 0.1 | -8.0 ± 0.1 | -3.7 ± 0.1 |

## 3.3. Ramped oxidation and incubation comparison

Figure 1 presents a comparison of the $^{14}$C content of the RO temperature fractions and respired $CO_2$ from the same soils during aerobic laboratory incubations (Houston et al. 2024b). These comparisons show that for each of the 0.5 cm depth samples, the $^{14}$C content of the respired $CO_2$ was greater than the $^{14}$C content of any of the RO temperature fractions in the same soils (Fig. 1). For the 5.5 cm depth samples, the $^{14}$C content of the $CO_2$ respired in the incubations was approximately equivalent to the $^{14}$C content of the 150-325°C RO temperature fraction (Fig. 1). For T2 15.5 cm, the $^{14}$C content of the respired $CO_2$ was also closest to the 150-325°C RO temperature fraction (Fig. 1). For the T1 18.5 cm and T3 19.5 cm samples, the $^{14}$C contents of the incubation $CO_2$ were depleted relative to the 150-325°C RO temperature fraction for both samples, and instead, were closest to the 325-425°C and 425-500°C RO temperature fractions, respectively (Fig. 1).

## 3.4. Activation Energy

Mean activation energy (µE) ranged from 157.50-170.97 kJ/mol for the 0.5 cm depth samples, 159.97-165.32 kJ/mol for the 5.5 cm depth samples, and 154.38-160.44 kJ/mol for the deepest samples (T1 18.5 cm, T2 15.5 cm, T3 19.5 cm. Table 3). The standard deviation of activation energy (σE) ranged from 23.16-35.83 kJ/mol for the 0.5 cm depth samples, 22.16-25.25 kJ/mol for the 5.5 cm depth samples, and 21.43-23.51 kJ/mol for the deepest samples (Table 3). Between the three depth increments, there were no significant changes in µE, σE, nor activation energy distribution (p (o,E) (Table 1. ANOVA; p = 0.47, 0.37, and 0.14, respectively).

*Table 3. Mean activation energy (μE), standard deviation of activation energy (σE), and activation*
*energy distribution for each sample.*

|  | μE (kJ/mol) | σE (kJ/mol) | p (σE) |
|---|---|---|---|
| **T1 0.5 cm** | 170.97 | 35.83 | 0.02 |
| **T1 5.5 cm** | 159.97 | 22.16 | 0.02 |
| **T1 18.5 cm** | 160.44 | 22.72 | 0.02 |
| **T2 0.5 cm** | 160.47 | 23.16 | 0.02 |
| **T2 5.5 cm** | 165.32 | 25.25 | 0.01 |
| **T2 15.5 cm** | 154.38 | 21.43 | 0.02 |
| **T3 0.5 cm** | 157.50 | 24.01 | 0.02 |
| **T3 5.5 cm** | 162.31 | 24.44 | 0.02 |
| **T3 19.5 cm** | 160.13 | 23.51 | 0.02 |


Table 4 shows μE and the associated σE for each thermal fraction. μE ranged from 131.04-133.23
kJ/mol for 150-325 °C, 156.83-157.78 kJ/mol for 325-425 °C, 176.14-177.79 kJ/mol for 425-500
°C, 185.44-199.19 kJ/mol for 500-650 °C, and 213.06-247.75 kJ/mol for 650-800 °C (Table 4). σE
ranged from 7.33-8.71 kJ/mol for 150-325 °C, 9.83-10.23 kJ/mol for 325-425 °C, 6.88-8.83 kJ/mol
for 425-500 °C, 3.68-16.04 kJ/mol for 500-650 °C, and 1.83-10.94 kJ/mol for 650-800 °C (Table 4).
μE and σE both varied significantly between the thermal fractions, increasing sequentially
(Kruskal-Wallis, $p = 0.001$ and 0.001, respectively). We therefore infer that the thermal
recalcitrance of RO fractions is greater at higher temperatures and use temperature as a proxy for
thermal reactivity herein.

*Table 4. Mean activation energy (μE) and standard deviation of activation energy (σE) for each RO temperature fraction for each sample.*

| | μE (σE) (kJ/mol) | | | | |
|---|---|---|---|---|---|
| | **150-325°C** | **325-425°C** | **425-500°C** | **500-650°C** | **650-800°C** |
| **T1 0.5 cm** | 132.43 (7.33) | 157.52 (10.17) | 177.79 (7.32) | 199.19 (16.04) | 242.42 (10.94) |
| **T1 5.5 cm** | 133.23 (8.07) | 157.00 (10.13) | 177.07 (7.58) | 191.06 (7.87) | 213.06 (2.12) |
| **T1 18.5 cm** | 131.90 (8.62) | 157.60 (9.83) | 176.93 (7.83) | 189.53 (6.70) | 239.73 (6.73) |
| **T2 0.5 cm** | 132.10 (8.21) | 157.42 (10.08) | 177.23 (7.38) | 191.39 (10.62) | 226.84 (6.81) |
| **T2 5.5 cm** | 132.15 (8.71) | 157.11 10.01) | 177.68 (8.83) | 195.80 (7.95) | 224.56 (4.55) |
| **T2 15.5 cm** | 131.04 (8.46) | 156.83 (10.23) | 176.69 (6.88) | 185.44 (3.68) | 247.74 (1.83) |
| **T3 0.5 cm** | 131.59 (7.48) | 157.33 (10.07) | 176.14 (7.07) | 193.65 (12.46) | 231.21 (10.04) |
| **T3 5.5 cm** | 133.05 (8.28) | 157.67 (10.14) | 177.19 (7.70) | 191.13 (9.11) | 236.57 (4.29) |
| **T3 19.5 cm** | 131.73 (8.38) | 157.78 (10.00) | 176.71 (7.34) | 191.7 (10.53) | 232.23 (9.69) |

## 4. Discussion

Soils are complex mixtures of many different OC sources and ages, with different vulnerabilities to decomposition and turnover. In this study, we aimed to improve our understanding of the carbon cycling of saltmarsh soils by measuring the $^{13}C$ and $^{14}C$ content of thermally-fractionated soil carbon pools, and comparing these results to the $^{14}C$ content of biologically evolved $CO_2$ from the same soils (Houston et al., 2024b).

### 4.1. Carbon provenance of ramped oxidation $CO_2$ fractions

The first three RO temperature fractions (150-325°C, 325-425°C, 425-500°C) were derived solely from OC sources, as IC begins to breakdown from ca. 550°C (Hemingway et al., 2017b). $CO_2$ from the 500-650°C and 650-800°C fractions may, however, have been evolved from a mix of OC and IC sources. The IC contents of the studied soils (0.11-0.48%) were low relative to OC contents (4.18-7.71%), and IC makes only 1.95-10.48% of the total soil C pool for these samples (Table A3). Wider μE ranges (mean activation energy of each thermal fraction) and increased bond strength

diversity (σE) compared to the first three RO fractions (Table 4) may have been caused by non-
first order decomposition of carbonates (a form of IC) from 550 °C, as first order decomposition
kinetics are a requirement for the *rampedpyrox* model (Hemingway et al., 2017b). Hemingway
(pers. comm. 16/01/2025) confirmed that due to the low amounts of carbonates in these samples
(Table A3) that it would be appropriate to calculate activation energies using the *rampedpyrox*
model.
IC could have been removed from our saltmarsh soil samples to allow complete analysis of the
soil OC pool, and many R(P)O studies have taken this approach (Bao et al., 2019b; Hemingway et
al., 2017b; Luk et al., 2021; Stoner et al., 2023; Williams and Rosenheim, 2015). However, our
samples have low IC contents (Table A3), and acid-treatment, which is required to remove IC from
samples, can cause losses of labile OC (Bao et al., 2019a). Indeed, in Hemingway et al. (2017),
acid treatment of samples prior to RO resulted in a shift of 4 % Modern $^{14}$C, which could change
one of our samples from having a pre-bomb $^{14}$C content to a post-bomb $^{14}$C content, or vice-
versa. A similar shift in $^{14}$C content for our samples could seriously impact the interpretations in
our study, and our ability to compare the $^{14}$C content of the $CO_2$ respired from bulk (untreated)
soils in the incubation experiments (Houston et al., 2024) to the $^{14}$C content of the RO fractions.
The soils in the incubation experiments were also not decarbonated as the acid-treatment would
have affected soil respiration processes and made the results incomparable to in-situ soil
degradation processes (Houston et al., 2024b).
**4.2. $^{14}$C content of ramped oxidation $CO_2$ fractions**
The $^{14}$C-RO content decreased over the four thermal fractions (150-325 °C, 325-425 °C, 425-500
°C, 500-650 °C. Fig. 1), implying that $^{14}$C-depleted OC had a greater thermal recalcitrance than
$^{14}$C-enriched OC for these saltmarsh soil samples. Since the $^{14}$C content of each RO fraction was
<100 % Modern (Table 1), each of the OC reactivity pools were likely to be predominantly
composed of carbon sequestered from the atmosphere before the 1963 $^{14}$C bomb-spike caused
by atmospheric nuclear weapons testing, although we cannot completely discount some
contributions from post-bomb carbon (Hajdas et al., 2021). Nevertheless, using $^{14}$C content as
an estimate of the age of the OC we can infer that the older ($^{14}$C-depleted) OC has a greater
thermal recalcitrance than young OC for these samples, which is consistent with previous
studies on the thermal reactivity of carbon stored in soils and sediments (e.g., Bao et al., 2019b;
Luk et al., 2021; Plante et al., 2013; Stoner et al., 2023).
The results suggest inhomogeneity within at least one of the temperature fractions for each
sample as, although there were no post-bomb $^{14}$C contents for the incubation or RO samples
(Table 1), there is likely to be a fraction of post-bomb (post-AD1955) OC in at least one of the
temperature fractions. Autochthonous OC sequestration (post-bomb) at this accreting saltmarsh
(Hajdas et al., 2021; Smeaton et al., 2024) may become obscured by contributions from pre-
bomb (pre-AD1955) OC. Observing the decline in $^{14}$C content with increasing temperature (Fig.
1), we hypothesise that, if present, this mixing of pre- and post-bomb C most likely occurred in
the 150-325°C fraction.
As the oldest (most $^{14}$C-depleted) C had the greatest thermal recalcitrance (Fig. 1), this
emphasises that saltmarshes accumulating greater amounts of older ($^{14}$C-depleted) OC will
likely provide the most thermally recalcitrant OC stores, and saltmarshes accumulating greater
proportions of contemporary OC, either through in-situ production or young allochthonous
components, contain soil OC stores which are of greater thermal lability (Komada et al., 2022;
Van de Broek et al., 2018). However, the $^{14}$C contents of the lowest temperature RO fraction (81-
98 % Modern; Table 1) highlight that although the thermal reactivity of OC decreases with $^{14}$C
content (Fig. 1), thermally labile OC can still be aged (at least hundreds of years old) for these
soils. Due to the often anaerobic and non-eroding conditions of buried sediments, saltmarshes
can therefore be stores of old, but thermally labile carbon. Of course, the thermal recalcitrance
of OC is not necessarily related to biological turnover time, as this is also dependent on the
prevailing environmental conditions (Schmidt et al., 2011; Spivak et al., 2019).

### 4.3. $^{13}$C content of ramped oxidation $CO_2$ fractions

$^{13}$C-RO increased sequentially with the thermal fractions (Fig. 2), due to greater contributions
from relatively $^{13}$C-enriched C sources from the higher temperature thermal fractions. The $^{13}$C-RO
contents of the 150-650 °C fractions were each typical of OC sources (Leng and Lewis, 2017),
whereas the $^{13}$C-RO contents of the 650-800 °C fraction were mostly typical of at least a partial
contribution from an IC source, with the exception of T2 5.5 cm and T3 5.5 cm (Table 2) (Brand et
al., 2014; Ramnarine et al., 2012). As IC can begin to evolve from 550 °C, it is possible that a mix
of OC and IC sources was present in the 500-650 °C thermal fractions.
As $^{13}$C-RO increased with temperature (Fig. 2, Table 2), $^{13}$C-enriched OC had a greater thermal
recalcitrance than $^{13}$C-depleted OC for these samples. Previous work has demonstrated that >80
% of the OC accumulating at Skinflats saltmarsh is autochthonous/terrestrial in origin (Miller et
al., 2023), with limited contributions from marine OC. The thermally recalcitrant OC was
potentially composed of a greater amount of OC which has undergone microbial decomposition
as this process tends to enrich the degraded OC in $^{13}$C (Boström et al., 2007; Etcheverría et al.,
2009; Luk et al., 2021; Sanderman and Grandy, 2020; Soldatova et al., 2024; Stoner et al., 2023).
The thermally recalcitrant OC may instead/also have been composed of more different OM
compounds (e.g., lignins, aromatics) than the more thermally labile OC (e.g., carbohydrates,
lipids) (Sanderman and Grandy, 2020). It is also possible that methodological artefacts, such as
kinetic fractionation, influenced the $^{13}C$-RO contents. Kinetic fractionation is explained by
different carbon isotopes evolving as $CO_2$ from the soil sample at different rates during the
ramped heating (Hemingway et al., 2017a). Kinetic fractionation would cause the $^{13}C$ content of
the evolved $CO_2$ to increase linearly with temperature (Hemingway et al., 2017a), and we cannot
rule out this artefact. Hemingway et al. (2017a) determined that kinetic fractionation was not an
important factor in their RPO procedure, but we used a different set-up (described in Garnett et
al., 2023).

**4.4. Changes in the isotopic content of ramped oxidation $CO_2$ fractions with depth**

The isotopic composition of the evolved $CO_2$ did not vary significantly with depth for any of the
temperature fractions. The lack of an increase in the age ($^{14}C$-depletion) of soil C with sample
depth is unusual, as typically C undergoes a burial process, and previous work has shown
diagenetic ageing of saltmarsh soils with depth as young OC is turned over faster than old OC
(Komada et al., 2022; Van de Broek et al., 2018).
Compared to other UK saltmarshes, Skinflats has relatively high C accumulation rates (Miller et
al., 2023; Smeaton et al., 2024). Depleted $^{14}C$ contents of the OC accumulating at the Skinflats
saltmarsh (Houston et al., 2024b) imply that a proportion of the OC being buried may already
have been aged at the time of deposition on the marsh surface, as the marsh formed in the 1930's
(Miller et al., 2023). The combination of high carbon accumulation rates and depleted soil $^{14}C$
contents implies that the Skinflats saltmarsh accumulates a high proportion of old, most likely
allochthonous OC. Some of the aged, allochthonous OC may have undergone significant
microbial processing and degradation prior to its accumulation in the saltmarsh soil. As the OM
is degraded, and the energetically favourable components are consumed, the resulting OM
becomes increasingly thermally recalcitrant (Luk et al., 2021; Sanderman and Grandy, 2020;
Soldatova et al., 2024). The accumulation of a high proportion of degraded OC on the Skinflats
saltmarsh may therefore explain the lack of observed change in the isotopic composition of the
soil OC pools with depth.
Not all old OC is degraded or thermally recalcitrant, and our results show that the Skinflats
saltmarsh is also a store of old ($^{14}C$-depleted), thermally labile OC (Fig. 1). Old OC can be
thermally labile if it 'ages' (is stored) in an environment with low decomposition rates, e.g., a
peatland (Dean et al., 2023), prior to transport and accumulation into the saltmarsh. There are
extensive peatlands in the Skinflats catchment, many of which are degrading (Lilly et al., 2012).
Regardless of the age and degradation state of the OC deposited onto the marsh surface, as it
gets buried it will undergo a degree of microbial processing and degradation in the saltmarsh soil
(Luk et al., 2021), but that process is potentially less prevalent at Skinflats than saltmarshes
accumulating younger, less degraded OC.
Through isotopic analysis of saltmarsh soils partitioned using ramped oxidation, we have
determined that increased thermal recalcitrance is related to older ([14]C-depleted; Fig. 1), more
degraded/microbially derived ([13]C-enriched; Fig. 2) soil C. These findings are consistent with
previous research on the thermal reactivity of soil and sediment C, that more energy is required
(higher temperature/μE) to decompose older ([14]C-depleted), degraded/microbially derived ([13]C-
enriched) C than younger ([14]C-enriched), less processed ([13]C-depleted) C (e.g., Bao et al., 2019b;
Plante et al., 2013; Stoner et al., 2023), including one saltmarsh study (Luk et al., 2021).

**4.5. Comparison of biologically and thermally evolved $CO_2$**

As the biological turnover time of OC depends on the prevailing environmental conditions as well
as thermal reactivity (Schmidt et al., 2011), the isotopic composition of the most biologically- and
thermally-reactive saltmarsh soil OC pools may not be the same. To determine if this is the case,
or not, we compared the isotopic composition of the RO thermal reactivity fractions to the
isotopic composition of the $CO_2$ that was evolved biologically during incubations of equivalent
samples (Houston et al., 2024b) (Fig. 1).
Figure 1 shows that for each of the 0.5 cm depth samples, the [14]C content of the $CO_2$ respired in
the aerobic laboratory experiments was [14]C-enriched relative to any of the RO temperature
fractions, which was also the case for the T3 5.5 cm sample (Table 3). The relative [14]C-enrichment
of the biologically respired $CO_2$ compared to the thermally evolved $CO_2$ was likely caused by
inhomogeneity in the OC thermal reactivity pools, as each defined thermal reactivity pool may be
composed of multiple OC sources of variable age and composition. As thermal recalcitrance is
related to [14]C-depletion for these samples (Fig. 1), we hypothesise that for saltmarsh soil samples
producing respired $CO_2$ that was [14]C-enriched relative to any of the RO fractions (T1 0.5 cm, T2
0.5 cm, T3 0.5 cm, T3 5.5 cm; Table 1, Fig. 1), that this $CO_2$ was biologically-produced from an OC
pool within the most thermally labile RO fraction (150-325°C). Thus, we suggest that even within
the 150-325 °C RO fraction there are pools of even younger OC, but that they are masked by older,
[14]C-depleted OC.
The [14]C content of respired $CO_2$ from the 5.5 cm depth samples tended to be closer to the [14]C
content of the lowest temperature (150-325°C) RO fraction (Fig. 1), implying that for these
samples the biologically evolved $CO_2$ was from a thermally labile OC pool. The T2 15.5 cm
respired $CO_2$ sample was also similar in [14]C content to the lowest temperature RO fraction,
whereas respired $CO_2$ from the slightly deeper T1 18.5 cm and T3 19.5 cm samples was [14]C-
depleted relative to the 150-325°C RO fraction, instead aligning closer to the higher temperature
RO fractions (Fig. 1). The biologically evolved $CO_2$ from T1 18.5 cm and T3 19.5 cm was therefore
not from a thermally labile OC pool. The [14]C content of the $CO_2$ evolved from the aerobic
incubations of T1 18.5 cm and T3 19.5 cm was hypothesized to have been derived from an
inorganic C source due to the enriched [13]C contents of -6.1‰ and -3.7‰, respectively (Houston
et al., 2024b). As IC biological turnover times are controlled by different factors than OC (Van
Dam et al., 2021), and the remainder of the samples were determined to evolve from OC
substrates, this is likely to explain why the [14]C content of the $CO_2$ evolved from the aerobic
incubation experiments for T1 18.5 cm and T3 19.5 cm did not align with the lowest temperature
(most thermally labile) RO fraction (Fig. 1). Therefore, there was a clear depth trend in the
relationship between the [14]C content of $CO_2$ respired in the aerobic incubation experiments and
the [14]C content of RO fractions of the same bulk soils. Degradation of the thermally labile OM
components to a more thermally recalcitrant state during burial may reduce the inhomogeneities
within the most thermally labile RO fraction for this study.
For seven out of nine samples (T1 18.5 cm and T3 19.5 cm being the outliers), the [14]C content of
the $CO_2$ evolved from the aerobic laboratory incubations was closest to the [14]C content of the
150-325°C RO temperature fraction. Therefore, even though the $CO_2$ evolved from the aerobic
incubation experiments was determined to be from a predominantly aged, allochthonous OC
source (Houston et al., 2024b), it can now also be shown to be derived from a predominantly
thermally labile OC pool (Fig. 1).
We did not attempt to relate the [13]C-RO to the [13]C content of the $CO_2$ respired in the incubation
experiments, due to the potential for microbial fractionation during the incubation experiments
which can change the [13]C content of the respired $CO_2$ and the resulting soil OC (Soldatova et al.,
2024; Werth and Kuzyakov, 2010). In contrast, [14]C results are normalised using the measured $\delta^{13}C$
values and are therefore immune to such isotopic fractionation effects.
**4.6. Implications**
Our results show that aged (presumed allochthonous), thermally labile OC stored in saltmarsh
soils remains vulnerable to loss to the atmosphere upon habitat drainage. Saltmarsh soils usually
exist in low-oxygen, tidally-inundated conditions which slow decomposition of OC (Chapman et
al., 2019), but many saltmarshes globally have been drained (and their soils subsequently

oxidised) to convert them for land uses such as housing developments and agriculture (Bromberg and Bertness, 2005; Campbell et al., 2022; Morris et al., 2012). In the Forth Estuary, where the Skinflats saltmarsh is located, as much as 50% of the intertidal area has been converted to agricultural land since 1600, often involving the drainage of saltmarshes (Hansom and McGlashan, 2008).

Protecting saltmarshes from degradation following drainage is listed as an eligible activity for generating carbon credits for blue carbon ecosystem (BCE) projects (VERRA, 2023) and there is significant potential for climate mitigation by avoided emissions from protecting vulnerable stocks of soil OC in BCEs (Goldstein et al., 2020; Griscom et al., 2017; Kwan et al., 2025; Sasmito et al., 2025). Similarly, the re-creation of saltmarsh habitat through managed realignment (rewetting by tidal inundation) of historic saltmarsh habitats which were previously reclaimed for land use purposes (e.g., agriculture) could reduce (and possible reverse) the emissions of aged OC to the atmosphere, both locally to Skinflats, and globally.

The evidence for the respiration of thermally labile, allochthonous OC from saltmarsh soils in a drainage degradation scenario demonstrates that at least this fraction of allochthonous OC should be counted as additional in carbon crediting projects and National GHG Inventories. Because allochthonous OC can account for up to 90 % of saltmarsh soil carbon (Komada et al., 2022), the inclusion of allochthonous OC (or even a fraction of it) would significantly increase the climate mitigation awarded to blue carbon projects (as carbon credits, or contributions to National GHG Inventories) (Houston et al., 2024a).

As the bioavailable OC respired in the experiments of Houston et al. (2024b) was (in most cases) from a predominantly thermally labile OC pool, and $^{14}$C-RO decreased (C became older) with increasing temperature (thermal recalcitrance), RO measurements could be useful for characterising the turnover times of OC pools for saltmarsh soils exposed to oxic conditions (drainage degradation scenario). The use of thermally defined OC pools to characterize OC turnover times for saltmarsh soils would require a modelling advancement to constrain degradation rates and residence times. Such efforts are not within the scope of this study but could inform additionality/permanence in these saltmarsh systems. Experimentally defined turnover times of OC thermal reactivity pools could, for example, provide a more robust approach than inclusion/exclusion of allochthonous OC from saltmarsh 'blue carbon' projects (Houston et al., 2024a).

Further research is needed to determine if the relationship between biological and thermal lability exists for different degradation scenarios such as nutrient enrichment, as OC turnover

time depends on the environmental conditions as well as the thermal lability of the OC pools.
Similarly, these experiments would need to be replicated for a wider range of saltmarshes (high
and low latitude saltmarshes, different typologies), as there are likely to be differences in OC
turnover in different systems.
The samples used for this study were from the low marsh zone only, but it is likely that the thermal
reactivity of the Skinflats saltmarsh soil C will vary spatially across the marsh, as the proportion
of OC sources has been shown to be variable across saltmarshes (Middelburg et al., 1997). Given
our findings that old ($^{14}$C-depleted) OC has greater thermal recalcitrance than young ($^{14}$C-
enriched) OC (Fig. 1), we anticipate that higher marsh zones, which typically have greater
proportions of autochthonous OC than lower marsh zones (Spohn et al., 2013), would contain a
greater proportion of thermally labile OC. However, it is important to recognise that some of the
young ($^{14}$C-enriched), autochthonous OC in saltmarsh soils can also be thermally recalcitrant.
As well as marsh zonation, we expect that the proportion of OC sources (and associated mix of
thermal reactivities) would also vary with proximity to marsh creeks which redistribute
autochthonous and allochthonous C across the saltmarsh habitat (Middelburg et al., 1997; Reed
et al., 1999). In previously published work we showed that Skinflats accumulates OC of a much
greater 'age' (depleted soil $^{14}$C contents) than two other saltmarshes in Scotland (Houston et al.,
2024b).
In this paper we have determined that age ($^{14}$C-content) is related to the thermal recalcitrance of
saltmarsh soil OC. We therefore speculate that sites accumulating younger OC would have more
thermally labile soil OC than sites accumulating older OC, like Skinflats, with wider implications
for the risks to these vulnerable stores of soil carbon from human disturbances.
**5. Conclusions**
This is the first study on saltmarsh soils to employ the ramped oxidation method. We show that
old ($^{14}$C-depleted) carbon dominates the thermally recalcitrant OC pools. The thermally labile OC
pools are also aged ($^{14}$C-depleted) compared to the contemporary atmosphere but are younger
than the thermally recalcitrant OC pools. These results highlight the role of saltmarshes as mixed
stores of both old, thermally recalcitrant OC, as well as younger, thermally labile OC.
We present the first comparison of the bioavailability ($CO_2$ evolved from incubation experiments;
Houston et al., 2024)) and the thermal reactivity (RO) of saltmarsh soil OC. We show that aged,
allochthonous $CO_2$ evolved from saltmarsh soils exposed to oxic conditions (Houston et al.,
2024b) are from a predominantly thermally labile OC pool. As saltmarsh soils exist mostly in low
oxygen, waterlogged conditions, management interventions to limit their exposure to elevated
oxygen availability may protect and conserve these stores of thermally labile OC and provide a
climate abatement service. Therefore, we recommend that thermally labile allochthonous OC
stored in saltmarsh soils should be counted as additional in some carbon crediting projects and
National GHG Inventories.
**Appendix A**
*Table A1. Additional $^{14}$C measurement from the 650-800 °C. $^{14}$C was measured at the Scottish*
*Universities Environmental Research Centre Accelerator Mass Spectrometer (AMS) Laboratory.*
*δ$^{13}$C (relative to Vienna PDB standard) was measured using isotope ratio mass spectrometry on*
*a Delta V (Thermo, Germany) and used to normalize the $^{14}$C results to a δ$^{13}$C = −25‰, which*
*were reported as %Modern $^{14}$C (i.e., Fraction modern × 100). Errors are reported to one*
*standard deviation from the mean.*

| Sample ID | % Modern $^{14}$C |
|---|---|
| Skin T1 0.5 cm 650-800 °C | 79.75 ± 0.50 |


*Table A2. Isotopic compositions measured by IRMS (δ$^{13}$C) compared to values estimated by the*
*rampedpyrox model (Hemingway, 2016). Modelled and measured δ$^{13}$C values are significantly*
*different (Mann-Whittney-U test, p = 0.04).*

| Sample ID | δ$^{13}$C (measured) | δ$^{13}$C (modelled) |
|---|---|---|
| Skin T1 0-1cm 150-325 °C | -24.7 ± 0.1 | -30.1 ± 0.2 |
| Skin T1 0-1cm 325-425 °C | -22.3 ± 0.1 | -27.8 ± 0.2 |
| Skin T1 0-1cm 425-500 °C | -20.2 ± 0.1 | -25.6 ± 0.2 |
| Skin T1 0-1cm 500-650 °C | -13.9 ± 0.1 | -19.5 ± 0.2 |
| Skin T1 0-1cm 650-800 °C | -5.6 ± 0.1 | -11.1 ± 0.2 |
| Skin T1 5-6cm 150-325 °C | -26.7 ± 0.1 | -27.7 ± 0.2 |
| Skin T1 5-6cm 325-425 °C | -24.7 ± 0.1 | -25.8 ± 0.2 |
| Skin T1 5-6cm 425-500 °C | -22.7 ± 0.1 | -23.8 ± 0.2 |
| Skin T1 5-6cm 500-650 °C | -21.3 ± 0.1 | -22.6 ± 0.2 |
| Skin T1 5-6cm 650-800 °C | -6.3 ± 0.1 | -8.0 ± 0.2 |
| Skin T1 18-19cm 150-325 °C | -25.9 ± 0.1 | -27.3 ± 0.2 |
| Skin T1 18-19cm 325-425 °C | -24.0 ± 0.1 | -25.4 ± 0.2 |
| Skin T1 18-19cm 425-500 °C | -22.6 ± 0.1 | -24.1 ± 0.2 |
| Skin T1 18-19cm 500-650 °C | -21.6 ± 0.1 | -23.3 ± 0.2 |
| Skin T1 18-19cm 650-800 °C | -9.5 ± 0.1 | -11.0 ± 0.2 |
| Skin T2 0-1cm 150-325 °C | -25.7 ± 0.1 | -27.9 ± 0.2 |
| Skin T2 0-1cm 325-425 °C | -23.6 ± 0.1 | -25.9 ± 0.2 |
| Skin T2 0-1cm 425-500 °C | -22.0 ± 0.1 | -24.3 ± 0.2 |
| Skin T2 0-1cm 500-650 °C | -19.9 ± 0.1 | -22.4 ± 0.2 |
| Skin T2 0-1cm 650-800 °C | -4.7 ± 0.1 | -7.2 ± 0.2 |
| Skin T2 5-6cm 150-325 °C | -26.5 ± 0.1 | -27.4 ± 0.2 |
| Skin T2 5-6cm 325-425 °C | -24.8 ± 0.1 | -25.8 ± 0.2 |
| Skin T2 5-6cm 425-500 °C | -23.0 ± 0.1 | -24.0 ± 0.2 |
| Skin T2 5-6cm 500-650 °C | -22.0 ± 0.1 | -23.1 ± 0.2 |

| | | |
|---|---|---|
| **Skin T2 5-6cm 650-800 °C** | -21.1 ± 0.1 | -22.3 ± 0.2 |
| **Skin T2 15-16cm 150-325 °C** | -28.0 ± 0.1 | -26.9 ± 0.2 |
| **Skin T2 15-16cm 325-425 °C** | -26.6 ± 0.1 | -25.5 ± 0.2 |
| **Skin Tr2 15-16cm 425-500 °C** | -25.4 ± 0.1 | -24.3 ± 0.2 |
| **Skin Tr2 15-16cm 500-650 °C** | -24.4 ± 0.1 | -23.6 ± 0.2 |
| **Skin T2 15-16cm 650-800 °C** | -4.0 ± 0.1 | -2.9 ± 0.2 |
| **Skin T3 0-1cm 150-325 °C** | -25.3 ± 0.1 | -27.3 ± 0.2 |
| **Skin T3 0-1cm 325-425 °C** | -23.7 ± 0.1 | -25.7 ± 0.2 |
| **Skin T3 0-1cm 425-500 °C** | -22.0 ± 0.1 | -24.1 ± 0.2 |
| **Skin T3 0-1cm 500-650 °C** | -18.7 ± 0.1 | -21.0 ± 0.2 |
| **Skin T3 0-1cm 650-800 °C** | -4.1 ± 0.1 | -6.2 ± 0.2 |
| **Skin T3 5-6cm 150-325 °C** | -26.2 ± 0.1 | -27.8 ± 0.2 |
| **Skin T3 5-6cm 325-425 °C** | -24.2 ± 0.1 | -25.9 ± 0.2 |
| **Skin T3 5-6cm 425-500 °C** | -22.6 ± 0.1 | -24.3 ± 0.2 |
| **Skin T3 5-6cm 500-650 °C** | -20.50 ± 0.1 | -22.32 ± 0.2 |
| **Skin T3 5-6cm 650-800 °C** | -12.60 ± 0.1 | -14.29 ± 0.2 |
| **Skin T3 19-20cm 150-325 °C** | -26.30 ± 0.1 | -27.48 ± 0.2 |
| **Skin T3 19-20cm 325-425 °C** | -24.20 ± 0.1 | -25.40 ± 0.2 |
| **Skin T3 19-20cm 425-500 °C** | -22.60 ± 0.1 | -23.78 ± 0.2 |
| **Skin T3 19-20cm 500-650 °C** | -21.20 ± 0.1 | -22.57 ± 0.2 |
| **Skin T3 19-20cm 650-800 °C** | -8.00 ± 0.1 | -9.30 ± 0.2 |


*Table A3. Soil carbon properties measured on equivalent sub-samples prior to the RO procedure, as reported in Houston et al. (2024). Total organic carbon (TOC), Total carbon (TC) for the soil samples were measured by a SoliTOC analyser (Elementar Analysensysteme, Hanau, Germany). $^{14}C$ was measured at the Scottish Universities Environmental Research Centre Accelerator Mass Spectrometer (AMS) Laboratory. $\delta^{13}C$ (relative to Vienna PDB standard) was measured using isotope ratio mass spectrometry on a Delta V (Thermo, Germany) and used to normalize the $^{14}C$ results to a $\delta^{13}C = -25‰$, which were reported as %Modern $^{14}C$ (i.e., Fraction modern × 100). Errors are reported to one standard deviation from the mean.*

| Sample ID | TOC (%) | TIC (%) | TC (%) | % Modern $^{14}C$ | $\delta^{13}C$ |
|---|---|---|---|---|---|
| **SK T1 0.5 cm** | 4.1 | 0.48 | 4.58 | 47.49 ± 0.23 | -23.5 ± 0.1 |
| **SK T1 5.5 cm** | 4.96 | 0.11 | 5.06 | 45.03 ± 0.20 | -24.5 ± 0.1 |
| **SK T1 18.5 cm** | 4.8 | 0.39 | 5.18 | 41.36 ± 0.19 | -23.8 ± 0.1 |
| **SK T2 0.5 cm** | 4.71 | 0.16 | 4.87 | 31.47 ± 0.15 | -22.2 ± 0.1 |
| **SK T2 5.5 cm** | 4.23 | 0.13 | 4.36 | 43.69 ± 0.21 | -24.1 ± 0.1 |
| **SK T2 15.5 cm** | 7.56 | 0.15 | 7.71 | 50.93 ± 0.24 | -25.1 ± 0.1 |
| **SK T3 0.5 cm** | 5.37 | 0.12 | 5.49 | 47.03 ± 0.22 | -23.7 ± 0.1 |
| **SK T3 5.5 cm** | 4.06 | 0.11 | 4.18 | 44.15 ± 0.21 | 24.0 ± 0.1 |
| **SK T3 19.5 cm** | 5.23 | 0.12 | 5.35 | 44.48 ± 0.21 | -24.1 ± 0.1 |

508

**Data Availability**

All data presented in this manuscript is available in the main text and appendices.

**Author Contribution Statement**

A.H. undertook the study, fieldwork, sample processing, data acquisition, and wrote the first draft of the manuscript. M.G. conducted the laboratory procedures with the help of A.H. A.H., W.A., and M.G. contributed to designing the study, fieldwork, and laboratory analyses. W.A., M.G., and J.S. oversaw the study and contributed to writing and revision of the manuscript.

**Competing Interests**

The authors declare that they have no conflict of interest.

**Acknowledgements**

We thank Jo Smith (University of Aberdeen) for her comments and edits on the first draft of this manuscript. We thank the NERC SUPER DTP for funding the PhD through which this research was undertaken (NE/S007342/1). We acknowledge support from the National Environmental Isotope Facility in funding the $^{14}$C measurements for this study under grant NE/S011587/1 (allocation numbers 2594.1022, 2709.1023). WENA also acknowledges support provided by the HORIZON-CL5-2023-D1-02-02 grant C-BLUES, Innovation to advance the evidence base for reporting of Blue Carbon inventories and greenhouse gas fluxes in coastal wetlands. Thanks are extended to Chloe Bates for assisting with sample collection. Finally, we thank the editor and both reviewers for their comments which have improved this manuscript.

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
