# Peer review of "Old Carbon, New Insights: Thermal Reactivity and Bioavailability of Saltmarsh Soils"

_EGUsphere, 2024_

## Referee Comment (RC1)

*Review of Houston et al. "Old Carbon, New Insights: Thermal Reactivity and Bioavailability of Saltmarsh Soils" (Biogeosciences; https://doi.org/10.5194/egusphere-2024-3281)*

Synopsis

The primary focus of this study is to determine Ramped Oxidation (RO) $^{14}$C activities (reported as "percent modern" or pMC) and $\delta^{13}$C values for organic carbon (OC) from a set of saltmarsh soil cores from the Skinflats saltmarsh in Scotland, UK. The authors find that, in general, pMC decreases and $\delta^{13}$C increases with increasing RO temperature in all studied samples. They interpret this result as evidence that saltmarshes store significant amounts of pre-aged, (thermally) recalcitrant OC (although I think the terms "labile" and "recalcitrant" get conflated with "bioavailable" throughout this study; detail below). The authors then compare these results with those of incubation experiments from the same set of cores (from an earlier study; Houston et al. 2024 *Limnol. Oceanogr.*), and conclude that, for most samples, respired OC matches most closely in its $^{14}$C activity with that from the lowest thermal fraction. This is used as evidence that remineralizing organisms largely use thermally labile OC as substrate.

This is a concise manuscript, and I think the comparison between RO and biological incubation $^{14}$C analysis is an interesting and underexplored area of research. I therefore find the overall theme of this manuscript interesting and fitting for *Biogeosciences*. That said, there are major weaknesses with the current manuscript that need to be addressed and fixed. I broadly define these as:

(i)      a lack of citation and acknowledgement of the primary RO literature, which is largely overlooked here;
(ii)     relatedly, poor framing of results within the existing RO data analysis and interpretation pipelines (i.e., no determination of activation energy, *E*, distributions);
(iii)    a lack of detail on measurement and sample analysis (most importantly, that samples were not decarbonated prior to RO);
(iv)    a lack of detail on methods validation and verification (e.g., measured bulk vs. RO mass-balance OC contents, $^{14}$C activities, and $\delta^{13}$C values);
(v)     conflation of concepts and overall "sloppy" use of terminology (particularly labile / recalcitrant vs. bioavailable, as well reporting of $^{14}$C ages for complex OC mixtures);
(vi)    improper use of regressions and data analysis (i.e., exponential and linear regressions when the *x* axis is not properly reported as a continuous function).

I detail each of these issues below. I refrain from making line-item comments, as a large portion of this text will likely need to be be re-written. I believe solving these issues---particularly issue (ii), in which the authors should utilize the well-established framework for interpreting RO data that has been developed over the past ~decade---will greatly improve the strengh of conclusions that can be drawn here. Only after these issues are addressed can I assess and comment on the discussions, interpretations, and conclusions

Thus, I do not support publication of the current version of this manuscript in *Biogeosciences*. However, I outline below my recommendations for how the authors could revise and re-write to better focus on more interesting discussion and interpretation within the context of the broader knowledge in this field. I would then be happy to re-review a revised version. Please do not hesitate to contact me regarding any questions on this review.

Sincerely,

Jordon Hemingway
jordon.hemingway@erdw.ethz.ch

**Major comments**

**1. Primary R(P)O literature and context**

There exists a rich literature of ramped (pyrolysis) oxidation studies dating back to Rosenheim et al. (2008) *Geochem. Geophys. Geosys.* that is largely ignored here. While I understand the primary focus of this manuscript is on saltmarsh soils, it is important to place this within the broader context thermal analyses, particularly when introducing the RO instrument (e.g., L76-85, L109-125). This body of literature consists, for example, of studies related to:

(i)     instrument design and underlying theory (e.g., Rosenheim et al. 2008 *Geochem. Geophys. Geosys.*; should be cited on L78 rather than Garnett et al. 2023);

(ii)    blank assessment (e.g., Hemingway et al. 2017 *Radiocarbon*, which is cited here but in a different context; Fernandez et al. 2014 *Anal. Chem.*, etc.);

(iii)   interpretive framework with respect to thermal activation energies and various OC sources (e.g., Hemingway et al. 2017 *Biogeosciences*, Hemingway et al. 2018 *Science*);

(iv)    data compilations, particularly as they relate to mechanisms of OC preservation such as organo-mineral interactions (e.g., Hemingway et al. 2019 *Nature*, Cui et al. 2022 *Science Advances*.

There is admittedly some self-citation in this list, but my overall point here is that many of the studies that first described and developed the framework for such thermal oxdation instrumentation is ignored in the present study.

**2. RO data analysis and interpretation**

Perhaps more importantly, the current manuscript takes an overly simplistic approach to data analysis and interpretation---particularly, the authors simply "bin" data into temperature windows (i.e., 150-325 °C, 325-425 °C, 425-500 °C, 500-650 °C, and 650-800 °C) and further bin these into "labile" (i.e., 150-425 °C) and "recalictrant" (i.e., 425-650 °C) fractions to perform all analyses and interpretations. This is not robust. This becomes apparently when the authors attempt to perform regressions using these "bins" on the $x$ axis despite the fact that they are not evenly distributed along temperature (see point 6, below).

Furthermore, the authors normalize all thermograms to maximum peak size within a given thermogram (i.e., such that the $y$ axis scales from 0 to 1, inclusive) and perform regression analyses on these normalized bins. This is again not robust since, for example, one sample could contain a tall-but-narrow peak that may not be a large contributor in terms of overall area (and thus fraction of total OC content). Rather, to be properly compared, all thermograms should be normalized such that the integral under each is equal to unity.

Forunately for the authors, there exists a well-established data anlaysis pipeline for the quantiative interpretation and comparison of RO data, as described in Hemingway et al. (2017) *Biogeoscineces* and easily implemented using the "rampedpyrox" Python package. In this framework, thermograms are converted to activation energy, $E$, distributions, $p(E)$, and RO $^{14}$C activities and $\delta^{13}$C values are plotted vs. the weighted-mean $E$ value for a given fraction. This approach allows for direct comparison between samples and datasets, and it has the added benefit of providing a continuous $x$ axis for interpreting isotopic trends with increasing thermal recalcitrance (e.g., as is attempted in the current manuscript's Figs. 2-3). Additionally, placing these data within an activation energy framework significantly simplifies and strengthens many discussion points, for example those made throughout Section 4.2. For example, one can determine thermogram bredth using $p(E)$ width, $\sigma_E$, which allows for easy comparison of the

importance of thermal recalcitrance between samples (as is currently done in a somewhat "clunky" manner beginning on L231).

I therefore strongly recommend that the authors interpret their data in an activation energy context, rather than as temperature "bins" as is currently done. Doing so will significantly improve data interpretation and will lead to more quantitative and robust trends.

**3. Analysis and measurement detail**

Significantly more detail on the analytical setup and sample preparation is necessary. For example, the reader should be able to easily determine: sample masses, carrier gas composition and flow rate (the authors state "stream of high purity oxygen" on L113, but I seriously doubt it is pure $O_2$), $CO_2$ masses needed for each $^{14}C$ and $^{13}C$ measurement, calculated isntrument blank contribution, etc. All of these details need to be listed and described in order for the reader to be able to trust and interpret any results.

Additionally, the reader has to wait until Section 4.3 (L296) before learning that samples have not be decarbonated prior to RO analysis! This is a major oversight, as this is critically important information for the reader to have when interpreting data. Relatedly, how do the authors account for the potential of Inorganic Carbon (IC) to begin to combust at ~550 °C, as has been shown in e.g., Hemingway et al. (2017) *Radiocarbon*? Such a phenomenon would lead to an OC/IC admixture at higher temperature fractions, which could be an alternative mechanism to explain increasing $\delta^{13}C$ value with increasing temperature (c.f., microbial decomposition, which is proposed as the mechanism on L284). Overall, the authors need to be much more clear about their sample handling procedures (particulalry lack of decarbonation), and they need to thoroughly and honestly discuss how these procedures may impact their results and interpretation. Currently, I see none of this in the manuscript.

Finally, a small point, but the authors state that IRMS-derived $\delta^{13}C$ data were, "used to normalise the $^{14}C$ results to a $\delta^{13}C$ of -25 ‰ to correct for isotopic fractionation" (L123). Is this really true? If so, that is a major break from typical procedure, which is to use the AMS-derived $^{13}C/^{12}C$ ratio for correction, as this includes any internal fractionation (e.g., during ionization).

**4. Data validation and verification**

Similar to the above point, I find several data validation and verification metrics missing. For example, what are the bulk sample %OC, %OC, pMC, and $\delta^{13}C$ values? How well do the authors' RO results reconstruct these bulk values? That is, if the authors take a weighted-average of their RO fractions, are they able to reproduce the bulk values within statistical uncertainty? These types of "sanity check" metrics are quite important and, without them, I find it very difficult to assess the robustness and validity of the reported RO data. Again, fortunately for the authors, there exists a well-established and -documented pipeline for performing these types of sanity checks, and it is easily implemented using the "rampedpyrox" python package.

Finally, another small(ish) point, but the authors suggest $^{13}C$ fractionation as a possible reason for increasing $\delta^{13}C$ values with increasing combustion temperatures (L289-291). We actually show in the cited paper that fractionation is *not* an important factor and likely only shifts results by ≤ 1 ‰ (Hemingway et al. 2017 *Radiocarbon*). Rather, the authors could consider mechanisms such as different macro-molecular compounds (e.g., lipids vs. carbohydrates) combusting at different temperature ranges or the importance of organo-mineral interactions for some compound classes. Furthermore, as mentioned above, the possibility of carbonate

contribution as low as ~550 °C needs to be addressed here, as this would result in a similar $\delta^{13}C$ trend for the medium- to high-temperature fractions.

**5. Concepts and terminology**
There are several instances throughout the manuscript where I find the terms "labile" and "recalcitrant" to be conflated with "bioavailable" (e.g., L62: "It is therefore assumed that old OC is mostly composed of recalcitrant (low reactivity) components, whereas young OC contains a greater proportion of labile (reactive) components; L78: "The energy required to thermally-evolve $CO_2$ is expected to be related to the energy required for biological degradation of OC, with $CO_2$ evolved at low temperatures deemed to be from more reactive soil OC pools than $CO_2$ evolved at higher temperatures"; L203: "implying that the reactivity of soil OC decreases with increasing temperature", L205: "low-temperature $CO_2$ peak as relatively 'labile' and the higher temperature $CO_2$ peak as 'recalcitrant' OC pools", etc.).

It is clear that the authors know the difference between these concepts, but the language of the current manuscript is sloppy in a way that that could easily lead to reader conflusion. I strongly suggest the authors only refer explicitly to *thermal* lability and *thermal* recalcitrance---as this is what their RO instrument is directly measuring. Then, any relationship to OC *bioavailability* or *turnover time* can be inferred or interpreted, particularly using $^{14}C$ activities. However, it is important to clearly articulate that increased *thermal recalictrance* need not correlate with biological turnover times---the former is merely an analytical tool to parse apart complex OC mixtures, while the latter is the true metric of interest in the environment.

Similarly, at some points in the manuscript the authors report their $^{14}C$ activities as traditional $^{14}C$ years before present (L265, L276) and interpret them within the context of paleoclimatic events (e.g., deglaciation). This is very dangerous. The $^{14}C$ age in years BP of *any complex OC mixture* is meaningless, as this is simply the weighted average of all compounds contained within this mixture. Thus, any correspondence between, say, the $^{14}C$ age of a given RO thermal fraction and the deglaciation is pure coincidence---it does not mean that all (or even any) OC compounds that are represented in said RO fraction were formed at that time. I strongly suggest the authors remove this interpretation.

**6. Regressions, plotting, and statistics**

I have several comments and suggestions related to the figures and interpretations thereof:

Fig. 1: I suggest using something besides color to distinguish $^{14}C$ activities---previous studies have included overlaid bar plots or scatter plots, which convey this information much more clearly. For example, when I printed the manuscript in black and white, I could not distinguish the $^{14}C$ colors at all. Also, looking at the thermogram, it is clearly evident that carbonate is present in the T1 0.5cm sample, but this is only mentioned much later in the manuscript (Section 4.2) and not at all addressed in the figure itself. Finally, strictly speaking, the gray region does not "indicate values outside of the $CO_2$ collection range (150-800 °C)", as the gray region also includes the range 650-800 °C for all but one sample.

Fig. 2 + Fig. 3: Here, the authors are plotting RO isotope results vs. an *x* axis that is equally spaced temperature fraction means despite the fact that said temperature fractions are *not* equally spaced. Thus, any regression functional forms have no meaning (i.e., these are not, in fact, exponential and linear, respectively, if the *x* axis were to show temperature in a proper, continuous form). This needs to be fixed. Again, fortunately for the authors, there exists an

entire interpretative framework based on thermal activation energy distributions, $p(E)$, as well as a python package that makes this possible with very few lines of code.

Additionally, by plotting all tempererature fractions as box-and-whisker plots, the reader loses significant information *for a given sample*. That is, there is no way to know, for example, which point in the 150-800 °C bin corresponds to the same sample in the 325-425 °C bin. By doing this, the authors are inherently reverting to the mean values across all samples for a given temperature bin, which loses nuance and information (and may be incorrect depending on how OC is distributed *whithin* each temperature fraction for different samples). I strongly suggest instead plotting each sample as a line in $^{14}C$ / $\delta^{13}C$ vs. temperature (or, better, activation energy) space so that the reader can follow trends for any given sample.

Additionally, for Fig. 3, the authors simply omit the 650-800 °C data. Only later in the discussion did I realize this is because the authors attribute these enriched values to carbonate contribution. However, there is no mention of this in the figure---a seemingly dishonest omission. I strongly suggest the authors include these data in the figure---they are valid data after all---and describe why they behave differently from the rest.

Finally, there are several instances where statements and interpretations seem to be in direct conflict (e.g., L132-135: "There were no significant trends with depth … Visually, for both T1 and T3 the size of the second major peak…"; similarly repeated on L210-211). Either the size of the peaks changes between samples, or it doesn't---if it is statistically insignificant, then any visual differences are moot and should not be interpreted. However, I suspect some of this statistical insnignificance is due to the particular method that the authors normalized their thermograms (see above comment).

---

## Author Response (AR1)

Dear Editor,

Thank you for your decision to reconsider our publication (egusphere-2024-3281) after major revisions. We greatly appreciated the comments of both reviewers, and we have taken the opportunity to take onboard these reviewer insights and believe that the manuscript is now greatly improved and hope that you find it suitable for publication.

In the following pages, we outline the reviewer comments and our responses. We hope that this will provide a transparent and clear account of all the revisions made. We also include a separate file with a cleaned final revised manuscript, and a tracked changes version.

In the following pages, line number references are for the final revised manuscript (clean version).

Please do not hesitate to contact me if you require any further information/clarifications.

Yours sincerely,

Alex Houston on behalf of the authors

**Reviewer 1**

We thank the reviewer for their review of our manuscript. We appreciate the constructive feedback and the opportunity to improve this manuscript. We welcome the comment that the topic of this manuscript is an interesting and underexplored area of research. In this response letter we will address the key points of the review, and outline how we adopted them into our revised manuscript. We note that due to the changes requested, much of our results and discussion have been rewritten.

**1. Primary R(P)O literature and context**

**There exists a rich literature of ramped (pyrolysis) oxidation studies dating back to Rosenheim et al. (2008) Geochem. Geophys. Geosys. that is largely ignored here. While I understand the primary focus of this manuscript is on saltmarsh soils, it is important to place this within the broader context thermal analyses, particularly when introducing the RO instrument (e.g., L76- 85, L109-125). This body of literature consists, for example, of studies related to:**

> **(i) instrument design and underlying theory (e.g., Rosenheim et al. 2008 Geochem. Geophys. Geosys.; should be cited on L78 rather than Garnett et al. 2023);**

> **(ii) blank assessment (e.g., Hemingway et al. 2017 Radiocarbon, which is cited here but in a different context; Fernandez et al. 2014 Anal. Chem., etc.);**

**(iii) interpretive framework with respect to thermal activation energies and various OC sources (e.g., Hemingway et al. 2017 Biogeosciences, Hemingway et al. 2018 Science);**

**(iv) data compilations, particularly as they relate to mechanisms of OC preservation such as organo-mineral interactions (e.g., Hemingway et al. 2019 Nature, Cui et al. 2022 Science Advances.**

**There is admittedly some self-citation in this list, but my overall point here is that many of the studies that first described and developed the framework for such thermal oxdation instrumentation is ignored in the present study.**

We agree with the reviewer's comment that there was insufficient citing of the primary ramped (pyrolysis) oxidation (R(P)O) literature in the original manuscript and thank them for sharing a list of potential sources to include. This issue stems from an attempt to keep the manuscript concise, however, we acknowledge that this has resulted in some of the primary RO literature being excluded or not covered in sufficient detail. We have covered these more extensively in the revised manuscript.

**L22-L56:** "*Ramped oxidation (RO) and ramped pyrolysis oxidation (RPO) have been used to estimate the thermal reactivity and biological turnover time of soil and sediment OC (Hemingway et al., 2017b; Plante et al., 2011; Rosenheim et al., 2008). RO and RPO involve measuring the quantity of $CO_2$ evolved as a sample is increasingly heated at a constant rate in an atmosphere containing oxygen (e.g., Plante et al., 2011; Stoner et al., 2023), or other gases, typically Helium (RPO: e.g., Hemingway et al., 2017a; Rosenheim et al., 2008). The temperature at which $CO_2$ is thermally-evolved is related to the activation energy required to thermally decompose C (Hemingway et al., 2017b), which is also an estimate of the energy required for biological degradation of OC (Peltre et al., 2013; Plante et al., 2013). $CO_2$ evolved at low temperatures is deemed to be from soil OC pools with a greater thermal lability than $CO_2$ evolved at higher temperatures (Peltre et al., 2013; Rosenheim et al., 2008). OC thermal reactivity pools can be examined by collecting the evolved $CO_2$ from set temperature ranges with distinct thermal reactivities and measuring the $^{14}C$ (age) and $^{13}C$ content (Rosenheim et al., 2008), which can then be related to the activation energy required to thermally decompose those C sources (Hemingway et al., 2017b).*

*The $^{14}C$ content of the thermal reactivity pools provides insight into the turnover time of each pool, with past research showing that the oldest soil organic matter (OM) (most depleted $^{14}C$ content) tends to dominate the most thermally recalcitrant fractions (Bao et al., 2019b; Plante et al., 2013; Stoner et al., 2023). Similar results have been found for saltmarsh soils (Luk et al., 2021). Young OC, which can be autochthonous or allochthonous (Van de Broek et al., 2018), has been found to turnover at a faster rate than old OC in saltmarsh soils (Komada et al., 2022; Van de Broek et al., 2018), implying that young OC may tend to be more thermally labile than old OC for saltmarsh soils.*

*The $^{13}C$ content of the thermal reactivity pools can also provide insight as to whether the source of OC has an influence on turnover time. Previous work has found that the $^{13}C$ content of evolved $CO_2$ tends to be more enriched at higher temperatures due to greater contributions from $^{13}C$-enriched, degraded/microbially derived OC (Luk et al., 2021; Sanderman and Grandy, 2020; Stoner et al., 2023). Similarly, comparisons of the isotopic composition of thermally-defined OC pools to their chemical properties have found that thermally labile OC is derived from mostly lipids and polysaccharides, whereas OC with a higher thermal recalcitrance is derived from a greater proportion of phenolic and aromatic compounds (Sanderman and Grandy, 2020). The thermal reactivity of soil and sediment OC is also influenced by the formation of organo-mineral complexes, which can physically and chemically stabilise OC (Bianchi et al., 2024; Hemingway*

*et al., 2019). Mineral-associations can increase the energy required for decomposition and have been found to increase thermal recalcitrance and to slow turnover times of soil and sediment OC (Hemingway et al., 2019; Stoner et al., 2023)."*

**2. RO data analysis and interpretation**

**Perhaps more importantly, the current manuscript takes an overly simplistic approach to data analysis and interpretation---particularly, the authors simply "bin" data into temperature windows (i.e., 150-325 °C, 325-425 °C, 425-500 °C, 500-650 °C, and 650-800 °C) and further bin these into "labile" (i.e., 150-425 °C) and "recalictrant" (i.e., 425-650 °C) fractions to perform all analyses and interpretations. This is not robust. This becomes apparently when the authors attempt to perform regressions using these "bins" on the x axis despite the fact that they are not evenly distributed along temperature (see point 6, below).**

**Furthermore, the authors normalize all thermograms to maximum peak size within a given thermogram (i.e., such that the y axis scales from 0 to 1, inclusive) and perform regression analyses on these normalized bins. This is again not robust since, for example, one sample could contain a tall-but-narrow peak that may not be a large contributor in terms of overall area (and thus fraction of total OC content). Rather, to be properly compared, all thermograms should be normalized such that the integral under each is equal to unity.**

**Forunately for the authors, there exists a well-established data anlaysis pipeline for the quantiative interpretation and comparison of RO data, as described in Hemingway et al. (2017) Biogeoscineces and easily implemented using the "rampedpyrox" Python package. In this framework, thermograms are converted to activation energy, E, distributions, p(E), and RO 14C activities and d13C values are plotted vs. the weighted-mean E value for a given fraction. This approach allows for direct comparison between samples and datasets, and it has the added benefit of providing a continuous x axis for interpreting isotopic trends with increasing thermal recalcitrance (e.g., as is attempted in the current manuscript's Figs. 2-3). Additionally, placing these data within an activation energy framework significantly simplifies and strengthens many discussion points, for example those made throughout Section 4.2. For example, one can determine thermogram bredth using p(E) width, sE, which allows for easy comparison of the 3 importance of thermal recalcitrance between samples (as is currently done in a somewhat "clunky" manner beginning on L231).**

**I therefore strongly recommend that the authors interpret their data in an activation energy context, rather than as temperature "bins" as is currently done. Doing so will significantly improve data interpretation and will lead to more quantitative and robust trends.**

We initially understood that the *rampedpyrox* model would not be suitable for our samples as they have not been decarbonated which is stated as a requirement (Hemingway et al., 2017b). Therefore, we proceeded with inferring thermal reactivity from temperature in the submitted manuscript.

Hemingway et al., 2017b; P3: "*the presence of carbonate will result in thermograms that cannot be accurately described by the model presented here, and we therefore argue in favor of acid treatment when using the RPO instrument to determine reaction energetics of carbonate-containing samples.*"

We agree with the reviewer's suggestion that the calculation of activation energies could improve the analysis and interpretation of data in this manuscript. Following receipt of this review we contacted the reviewer to enquire if the '*rampedpyrox*' model was suitable for our samples, after all. Despite the published guidance (Hemingway et al., 2017b), the reviewer has confirmed that the method is suitable because of the low amount of carbonates in our samples. We have

therefore implemented the '*rampedpyrox*' model for analysis and presentation of the thermograms and in the revised manuscript. We do not use the 'rampedpyrox' model for presentation of our isotope data as it applies a blank correction to $^{14}$C which is not relevant to the analytical set-up for this study (Garnett et al., 2023), and the $^{13}$C values generated varied significantly from our IRMS measured values (these are presented in Table A2 in the appendix to the revised manuscript).

**L157-167:** "*Continuous activation energy distributions were modelled from thermograms using the 'rampedpyrox' package in Python V3.8 (Hemingway, 2016; Hemingway et al., 2017b). The rampedpyrox model calculates mean activation energies (µE) and the standard deviation of activation energy (σE), which is a measure of the heterogeneity of bond strength, for each temperature fraction which $CO_2$ was collected from. Mean µE, σE and activation energy distribution (p (o,E)) are also calculated for each sample using the rampedpyrox model. We do not use the rampedpyrox model for calculation of isotope values as it applies a blank correction to $^{14}$C (Hemingway et al., 2017a, b) which is not relevant to the analytical set-up for this study (Garnett et al., 2023), and the $^{13}$C values generated varied significantly from our IRMS measured values (Table A2). Further data analysis and visualisation of thermograms and isotopic data was undertaken using RStudio V4.2.2 (R Core Team, 2022).*"

Regarding the normalisation of thermograms to maximum peak size, this is simply to aid visual comparison of the curves between the samples. The regressions were not based on the normalised curves, but on the volume of $CO_2$ evolved over the defined temperature intervals. We retain this approach in the updated Figures 1 (**L170**) and 2 (**L194**) in the revised manuscript.

The figures used in the results section of the original manuscript have been updated in the revised manuscript. The regressions presented in the initial manuscript have been removed from the revised version.

Due to the change in our data analysis approach, much of the results (**L168-239**) and discussion (**L240-472**) sections have been rewritten.

**3. Analysis and measurement detail**

**Significantly more detail on the analytical setup and sample preparation is necessary. For example, the reader should be able to easily determine: sample masses, carrier gas composition and flow rate (the authors state "stream of high purity oxygen" on L113, but I seriously doubt it is pure O2), CO2 masses needed for each 14C and 13C measurement, calculated isntrument blank contribution, etc. All of these details need to be listed and described in order for the reader to be able to trust and interpret any results.**

A published paper verifying the RO methodology used in this manuscript already exists (Garnett et al., 2023) and is cited in the original submission of this manuscript. We have added the required details such as carrier gas composition, sample masses, and flow rate to the methods section of the revised manuscript.

**L101-109:** "*The samples were sent to the NEIF Radiocarbon Laboratory for RO, which is described in Garnett et al. (2023). The RO procedure involved two stages, a first combustion to determine the relationship between the rate of $CO_2$ evolution and temperature (thermogram), and a second combustion where evolved sample gases were collected across defined temperature ranges, for subsequent isotope analysis. For the first combustion, ca. 200 mg of dried and homogenized sample material was weighed into a quartz vial which was inset into a quartz combustion tube, which was subsequently placed into a furnace set initially to room temperature. The furnace was progressively heated at a constant rate of 5°C per minute to 800°C in a stream of high purity oxygen (N5.5, BOC, UK).*"

We have also added detail on the required $CO_2$ mass for isotope measurement to the revised manuscript.

**L121-128:** "*For each sample, the required mass of material to evolve sufficient $CO_2$ (> 3 mL) for $^{14}C$ measurement was calculated based on the thermogram. A new sub-set from the original dried and homogenised sample was then re-run following the RO procedure outlined above, but instead of venting to atmosphere, after its measurement the evolved $CO_2$ was collected into foil gas bags based on the defined temperature ranges. $CO_2$ was collected for $^{13}C$ analysis from 650-800 °C, but sufficient $CO_2$ was evolved for $^{14}C$ analysis from this thermal fraction for only one sample (T1 0.5 cm, Table A1) and we do not consider this fraction further because it is likely dominated by carbonates and not relevant to the purpose of this study.*"

**Additionally, the reader has to wait until Section 4.3 (L296) before learning that samples have not be decarbonated prior to RO analysis! This is a major oversight, as this is critically important information for the reader to have when interpreting data. Relatedly, how do the authors account for the potential of Inorganic Carbon (IC) to begin to combust at ~550 °C, as has been shown in e.g., Hemingway et al. (2017) Radiocarbon? Such a phenomenon would lead to an OC/IC admixture at higher temperature fractions, which could be an alternative mechanism to explain increasing δ13 C value with increasing temperature (c.f., microbial decomposition, which is proposed as the mechanism on L284). Overall, the authors need to be much more clear about their sample handling procedures (particulalry lack of decarbonation), and they need to thoroughly and honestly discuss how these procedures may impact their results and interpretation. Currently, I see none of this in the manuscript.**

The decision to not remove carbonates from our samples was taken to avoid the potential loss of carbon from labile fractions during acid treatment (Bao et al., 2019). Importantly, we note that in Hemingway et al. (2017b), acid treatment of samples prior to RO resulted in a shift of 0.04 Fm $^{14}C$. This could shift a sample from having a pre-bomb $^{14}C$ content to a post-bomb $^{14}C$ content, or vice-versa. A similar shift in $^{14}C$ content for our samples could seriously impact the interpretations in our study, and our ability to compare the $^{14}C$ content of the $CO_2$ respired from the incubation experiments (Houston et al., 2024) to the $^{14}C$ content of the $CO_2$ evolved during the RO procedure. This is a crucial component of this work.

We have added detail on the treatment of these samples to the methods section of the revised manuscript to avoid confusing future readers.

**L94-100:** "*Unlike most RO and RPO studies (e.g., Hemingway et al., 2017b), we did not remove carbonates from our samples. Acid treatment, which is required to remove carbonates from samples has been demonstrated to result in losses from the labile OC fraction (Bao et al., 2019a). A loss of labile OC for our samples could seriously impact the interpretations in our study, and our ability to compare the $^{14}C$ content of the $CO_2$ respired from bulk (untreated) soils in the incubation experiments (Houston et al., 2024b) to the $^{14}C$ content of the RO thermal fractions.*"

We have also reported the inorganic carbon (IC) content of the bulk soil samples in the revised supplementary information (Table A3) and added detail on the potential for IC to be evolved as $CO_2$ from 550 °C to our discussion section in the revised manuscript.

**L247-271:** "*The first three RO temperature fractions (150-325°C, 325-425°C, 425-500°C) were derived solely from OC sources, as IC begins to breakdown from ca. 550°C (Hemingway et al., 2017b). $CO_2$ from the 500-650°C and 650-800°C fractions may, however, have been evolved from a mix of OC and IC sources. The IC contents of the studied soils (0.11-0.48%) were low relative to OC contents (4.18-7.71%), and IC makes only 1.95-10.48% of the total soil C pool for these samples (Table A3). Wider µE ranges (mean activation energy of each thermal fraction) and*

*increased bond strength diversity (σE) compared to the first three RO fractions (Table 4) may have been caused by non-first order decomposition of carbonates (a form of IC) from 550 °C, as first order decomposition kinetics are a requirement for the rampedpyrox model (Hemingway et al., 2017b). Hemingway (pers. comm. 16/01/2025) confirmed that due to the low amounts of carbonates in these samples (Table A3) that it would be appropriate to calculate activation energies using the rampedpyrox model.*

*IC could have been removed from our saltmarsh soil samples to allow complete analysis of the soil OC pool, and many R(P)O studies have taken this approach (Bao et al., 2019b; Hemingway et al., 2017b; Luk et al., 2021; Stoner et al., 2023; Williams and Rosenheim, 2015). However, our samples have low IC contents (Table A3), and acid-treatment, which is required to remove IC from samples, can cause losses of labile OC (Bao et al., 2019a). Indeed, in Hemingway et al. (2017), acid treatment of samples prior to RO resulted in a shift of 4 % Modern $^{14}$C, which could change one of our samples from having a pre-bomb $^{14}$C content to a post-bomb $^{14}$C content, or vice-versa. A similar shift in $^{14}$C content for our samples could seriously impact the interpretations in our study, and our ability to compare the $^{14}$C content of the $CO_2$ respired from bulk (untreated) soils in the incubation experiments (Houston et al., 2024) to the $^{14}$C content of the RO fractions. The soils in the incubation experiments were also not decarbonated as the acid-treatment would have affected soil respiration processes and made the results incomparable to in-situ soil degradation processes (Houston et al., 2024b)."*

**Finally, a small point, but the authors state that IRMS-derived δ13C data were, "used to normalise the 14C results to a δ13C of -25 ‰ to correct for isotopic fractionation" (L123). Is this really true? If so, that is a major break from typical procedure, which is to use the AMS-derived 13C/ 12 C ratio for correction, as this includes any internal fractionation (e.g., during ionization).**

It is incorrect to state that using $\delta^{13}$C values (determined using isotope ratio mass spectrometry (IRMS) from an aliquot of the sample $CO_2$) to normalise $^{14}$C results represents a major break from typical procedure. Use of IRMS $\delta^{13}$C values for normalising AMS data is clearly supported in several key AMS methods papers (Donahue et al., 1990; McNichol et al., 2001) and while online AMS $^{13}$C measurements can be advantageous particularly for small samples (Santos et al., 2007), many labs employ IRMS values for normalisation (McIntyre et al., 2017) and quality assurance results show that the method is clearly reliable (Ascough et al., 2024).

**4. Data validation and verification**

**Similar to the above point, I find several data validation and verification metrics missing. For example, what are the bulk sample %OC, %OC, pMC, and δ13C values? How well do the authors' RO results reconstruct these bulk values? That is, if the authors take a weightedaverage of their RO fractions, are they able to reproduce the bulk values within statistical uncertainty? These types of "sanity check" metrics are quite important and, without them, I find it very difficult to assess the robustness and validity of the reported RO data. Again, fortunately for the authors, there exists a well-established and -documented pipeline for performing these types of sanity checks, and it is easily implemented using the "rampedpyrox" python package.**

The RO samples are sub-sets of those analysed in Houston et al. (2024), where the bulk sample %OC, %IC, pMC, and δ13C values are all reported (Houston et al., 2024). To aid comparison for readers, we have added these into the supplementary material in the revised manuscript (Table A3).

We cannot calculate mass balances for the RO and bulk soil isotopic contents, as the latter were acid-treated, while the former were not, so any comparisons would not be robust.

As stated earlier in this response letter, we do not use the *rampedpyrox* model for our isotope data.

**L162-166:** "*We do not use the rampedpyrox model for calculation of isotope values as it applies a blank correction to $^{14}C$ (Hemingway et al., 2017a, b) which is not relevant to the analytical set-up for this study (Garnett et al., 2023), and the $^{13}C$ values generated varied significantly from our IRMS measured values (Table A2).*"

**Finally, another small(ish) point, but the authors suggest 13C fractionation as a possible reason for increasing δ13C values with increasing combustion temperatures (L289-291). We actually show in the cited paper that fractionation is not an important factor and likely only shifts results by ≤ 1 ‰ (Hemingway et al. 2017 Radiocarbon). Rather, the authors could consider mechanisms such as different macro-molecular compounds (e.g., lipids vs. carbohydrates) combusting at different temperature ranges or the importance of organo-mineral interactions for some compound classes. Furthermore, as mentioned above, the possibility of carbonate contribution as low as ~550 °C needs to be addressed here, as this would result in a similar δ13C trend for the medium- to high-temperature fractions.**

We welcome the reviewer's point about the potential (or lack thereof) for $^{13}C$ fractionation during the RO procedure. The isotopic fractionation correction by the '*rampedpyrox*' model corrected our $δ^{13}C$ signatures by 1-6 ‰, so we used our IRMS measured signatures instead, which we note in the new 'data analysis' section of the revised manuscript.

**L165-169:** "*We do not use the rampedpyrox model for calculation of isotope values as it applies a blank correction to $^{14}C$ (Hemingway et al., 2017a, b) which is not relevant to the analytical set-up for this study (Garnett et al., 2023), and the $^{13}C$ values generated varied significantly from our IRMS measured values (Table A2).*"

We have added more detail to our discussion for the potential for different C sources to combust at different temperatures (including IC). However, $^{14}C$ is the main focus of this manuscript (comparison of $^{14}C$ content of respired $CO_2$ from incubation experiments and $^{14}C$ content of thermal reactivity fractions).

**L305-328:** "*$^{13}C$-RO increased sequentially with the thermal fractions (Fig. 2), due to greater contributions from relatively $^{13}C$-enriched C sources from the higher temperature thermal fractions. The $^{13}C$-RO contents of the 150-650 °C fractions were each typical of OC sources (Leng and Lewis, 2017), whereas the $^{13}C$-RO contents of the 650-800 °C fraction were mostly typical of at least a partial contribution from an IC source, with the exception of T2 5.5 cm and T3 5.5 cm (Table 2) (Brand et al., 2014; Ramnarine et al., 2012). As IC can begin to evolve from 550 °C, it is possible that a mix of OC and IC sources was present in the 500-650 °C thermal fractions.*

*As $^{13}C$-RO increased with temperature (Fig. 2, Table 2), $^{13}C$-enriched OC had a greater thermal recalcitrance than $^{13}C$-depleted OC for these samples. Previous work has demonstrated that >80 % of the OC accumulating at Skinflats saltmarsh is autochthonous/terrestrial in origin (Miller et al., 2023), with limited contributions from marine OC. The thermally recalcitrant OC was potentially composed of a greater amount of OC which has undergone microbial decomposition as this process tends to enrich the degraded OC in $^{13}C$ (Boström et al., 2007; Etcheverría et al., 2009; Luk et al., 2021; Sanderman and Grandy, 2020; Soldatova et al., 2024; Stoner et al., 2023). The thermally recalcitrant OC may instead/also have been composed of more different OM compounds (e.g., lignins, aromatics) than the more thermally labile OC (e.g., carbohydrates, lipids) (Sanderman and Grandy, 2020). It is also possible that methodological artefacts, such as kinetic fractionation, influenced the $^{13}C$-RO contents. Kinetic fractionation is explained by different carbon isotopes evolving as $CO_2$ from the soil sample at different rates during the ramped heating (Hemingway et al., 2017a). Kinetic fractionation would cause the $^{13}C$ content of*

*the evolved $CO_2$ to increase linearly with temperature (Hemingway et al., 2017a), and we cannot rule out this artefact. Hemingway et al. (2017a) determined that kinetic fractionation was not an important factor in their RPO procedure, but we used a different set-up (described in Garnett et al., 2023)."*

**5. Concepts and terminology**

**There are several instances throughout the manuscript where I find the terms "labile" and "recalcitrant" to be conflated with "bioavailable" (e.g., L62: "It is therefore assumed that old OC is mostly composed of recalcitrant (low reactivity) components, whereas young OC contains a greater proportion of labile (reactive) components; L78: "The energy required to thermally-evolve CO2 is expected to be related to the energy required for biological degradation of OC, with CO2 evolved at low temperatures deemed to be from more reactive soil OC pools than CO2 evolved at higher temperatures"; L203: "implying that the reactivity of soil OC decreases with increasing temperature", L205: "low-temperature CO2 peak as relatively 'labile' and the higher temperature CO2 peak as 'recalcitrant' OC pools", etc.).**

**It is clear that the authors know the difference between these concepts, but the language of the current manuscript is sloppy in a way that that could easily lead to reader conflusion. I strongly suggest the authors only refer explicitly to thermal lability and thermal recalcitrance---as this is what their RO instrument is directly measuring. Then, any relationship to OC bioavailability or turnover time can be inferred or interpreted, particularly using 14C activities. However, it is important to clearly articulate that increased thermal recalictrance need not correlate with biological turnover times---the former is merely an analytical tool to parse apart complex OC mixtures, while the latter is the true metric of interest in the environment.**

We have improved our terminology usage in the revised manuscript and thank the reviewer for their suggested terms of "*thermal recalcitrance*" and "*thermal lability*" which we have used when referring to the thermal reactivity of the OC pools. We have kept the use of "*bioavailable OC*" when referring to the $CO_2$ respired from the incubation experiments.

We agree with the reviewer that it is important to highlight to the reader that thermal reactivity does not have to correlate with biological turnover times. This was one of the motivations for undertaking this work as much previous work has been undertaken on the 'reactivity' of soil/sediment OC but there has been limited work comparing this to 'bioavailable' carbon under set environmental conditions.

**L365-370:** "*As the biological turnover time of OC depends on the prevailing environmental conditions as well as thermal reactivity (Schmidt et al., 2011), the isotopic composition of the most biologically- and thermally-reactive saltmarsh soil OC pools may not be the same. To determine if this is the case, or not, we compared the isotopic composition of the RO thermal reactivity fractions to the isotopic composition of the $CO_2$ that was evolved biologically during incubations of equivalent samples (Houston et al., 2024b) (Fig. 1).*"

**Similarly, at some points in the manuscript the authors report their 14C activities as traditional 14C years before present (L265, L276) and interpret them within the context of paleoclimatic events (e.g., deglaciation). This is very dangerous. The 14C age in years BP of any complex OC mixture is meaningless, as this is simply the weighted average of all compounds contained within this mixture. Thus, any correspondence between, say, the 14C age of a given RO thermal fraction and the deglaciation is pure coincidence---it does not mean that all (or even any) OC compounds that are represented in said RO fraction were formed at that time. I strongly suggest the authors remove this interpretation.**

We take on board the reviewer's comments on not using [14]C 'ages'. We had included this to provide context to the most [14]C-depleted samples and to highlight the potential for [14]C-dead contributions but that they aren't necessary to achieve these low [14]C contents for the region. However, we have removed this interpretation from the revised manuscript to avoid inappropriate reporting of complex OC mixtures.

**I have several comments and suggestions related to the figures and interpretations thereof:**

We have replaced each of the figures from the original manuscript in the revised version based on your comments below, and the revised data analysis.

**Fig. 1: I suggest using something besides color to distinguish 14C activities---previous studies have included overlaid bar plots or scatter plots, which convey this information much more clearly. For example, when I printed the manuscript in black and white, I could not distinguish the 14C colors at all. Also, looking at the thermogram, it is clearly evident that carbonate is present in the T1 0.5cm sample, but this is only mentioned much later in the manuscript (Section 4.2) and not at all addressed in the figure itself. Finally, strictly speaking, the gray region does not "indicate values outside of the CO2 collection range (150-800 °C)", as the gray region also includes the range 650-800 °C for all but one sample.**

We have replaced Figure 1 with a new plot (also Figure 1, **L170**) showing the thermograms as red lines normalised to peak $CO_2$. The [14]C content of the RO thermal fractions is displayed as bars, and the [14]C content of the $CO_2$ respired from the incubation experiments as dashed green lines. We believe that this is a much clearer approach. In the revised manuscript, we also have Figure 2 (**L193**) which shows the same information but for RO-[13]C.

**Fig. 2 + Fig. 3: Here, the authors are plotting RO isotope results vs. an x axis that is equally spaced temperature fraction means despite the fact that said temperature fractions are not equally spaced. Thus, any regression functional forms have no meaning (i.e., these are not, in fact, exponential and linear, respectively, if the x axis were to show temperature in a proper, continuous form). This needs to be fixed. Again, fortunately for the authors, there exists an 5 entire interpretative framework based on thermal activation energy distributions, p(E), as well as a python package that makes this possible with very few lines of code.**

In the revised manuscript, we have replaced all figures and removed all regressions. As stated earlier in this reply, we do not compare isotope data to activation energies.

**Additionally, by plotting all tempererature fractions as box-and-whisker plots, the reader loses significant information for a given sample. That is, there is no way to know, for example, which point in the 150-800 °C bin corresponds to the same sample in the 325-425 °C bin. By doing this, the authors are inherently reverting to the mean values across all samples for a given temperature bin, which loses nuance and information (and may be incorrect depending on how OC is distributed whithin each temperature fraction for different samples). I strongly suggest instead plotting each sample as a line in 14C / δ13C vs. temperature (or, better, activation energy) space so that the reader can follow trends for any given sample.**

Figures 1 and 2 (see above) in the revised manuscript plot the isotope data against temperature for each sample individually.

**Additionally, for Fig. 3, the authors simply omit the 650-800 °C data. Only later in the discussion did I realize this is because the authors attribute these enriched values to carbonate contribution. However, there is no mention of this in the figure---a seemingly dishonest omission. I strongly suggest the authors include these data in the figure---they are valid data after all---and describe why they behave differently from the rest.**

The omission of the 650-800 °C RO fraction was to compare the OC pools. However, we strongly disagree with the reviewer that this was a dishonest omission as the $\delta^{13}C$ for the 650-800 °C RO fraction were reported in Table 2 of the original manuscript, directly below this figure.

This accusation of dishonesty, whether "seemingly dishonest" or not, is in direct conflict with this journal's code of conduct for reviewers, which state, "reviewers…should never include personal criticism of an author in a manuscript review." We therefore object to the use of this language in the review.

In the revised manuscript, we plot all five RO thermal fractions for $^{13}C$ analysis (Figure 2) and the first four temperatures for $^{14}C$ analysis (Figure 1). We only have one $^{14}C$ measurement from the 650-800 °C RO fraction which we report in the appendix (Table A1), as it is not robust to gain insights from only one sample.

**Finally, there are several instances where statements and interpretations seem to be in direct conflict (e.g., L132-135: "There were no significant trends with depth … Visually, for both T1 and T3 the size of the second major peak…"; similarly repeated on L210-211). Either the size of the peaks changes between samples, or it doesn't---if it is statistically insignificant, then any visual differences are moot and should not be interpreted. However, I suspect some of this statistical insnignificance is due to the particular method that the authors normalized their thermograms (see above comment).**

We have removed all discussion of statistically insignificant trends from the revised manuscript and focus our discussion on comparison of the isotope contents of the RO thermal fractions.

**Reviewer 2**

We thank Reviewer 2 for their constructive and helpful comments on our manuscript, and for their highlighting of the importance and novelty of this research:

*"This is an important piece of work investigating the reactivity of saltmarsh soil organic carbon. The study found that 14C-depleted (older) carbon evolved from higher temperature ramped oxidation fractions, indicating that older carbon dominates the thermally recalcitrant fractions. The work does progress the field and offers new insights into our understanding of the composition of this important carbon sink."*

Below, we outline our response to each of their specific comments raised.

**The use of the term "pre-aged" (especially in the abstract) is ambiguous and should be clarified to the reader without needing to go back to Houston et al (2024).**

In the interpretation of $^{14}C$ measurements, 'age' usually reflects when the carbon is fixed/isolated from the atmosphere, so all OC stored in saltmarsh soils can be considered 'aged' compared to the contemporary atmosphere. In the revised manuscript we now use 'aged' to refer to OC depleted compared to the contemporary atmosphere.

**In the methods section there should be a sub section on data analysis. What programmes were used, how was the data treated?**

We have added a 'data analysis' section to the revised manuscript. In the original version of this manuscript all data analysis and visualisation of thermograms and isotopic data was undertaken using RStudio V4.2.2. In the revised manuscript, we have used Python V3.8 to implement the '*rampedpyrox*' package (Hemingway, 2016; Hemingway et al., 2017b) to calculate activation energy distributions for our samples from the time-temperature thermograms, as requested by Reviewer 1.

**L157-167:** *"Continuous activation energy distributions were modelled from thermograms using the 'rampedpyrox' package in Python V3.8 (Hemingway, 2016; Hemingway et al., 2017b). The rampedpyrox model calculates mean activation energies ($\mu E$) and the standard deviation of activation energy ($\sigma E$), which is a measure of the heterogeneity of bond strength, for each temperature fraction which $CO_2$ was collected from. Mean $\mu E$, $\sigma E$ and activation energy distribution (p (o,E)) are also calculated for each sample using the rampedpyrox model. We do not use the rampedpyrox model for calculation of isotope values as it applies a blank correction to $^{14}C$ (Hemingway et al., 2017a, b) which is not relevant to the analytical set-up for this study (Garnett et al., 2023), and the $^{13}C$ values generated varied significantly from our IRMS measured values (Table A2). Further data analysis and visualisation of thermograms and isotopic data was undertaken using RStudio V4.2.2 (R Core Team, 2022)."*

**Also Line 105-106. "the deepest sample from each core being the deepest retrieved sample" – does this mean that the depth to refusal at this site is 20cm?**

No, in this study we used a golf-hole corer and it's length is 20 cm. The depth to refusal at this site is >20 cm (see Miller et al., 2023), and variable across the site. In the revised manuscript we have added an explicit statement about the length of the coring device at the appropriate point in the text and this adds the necessary clarification.

**L85-89:** *"Briefly, the cores were split into 1 cm thick slices as follows: core T1 (0-1 cm, 5-6 cm, and 18-19 cm); T2 (0-1 cm, 5-6 cm, and 15-16 cm), and T3 (0-1 cm, 5-6 cm, and 19-20 cm) (with the deepest sample from each core being the deepest retrieved sample from the 20 cm length of the corer. On the occasions when a full core was not retrieved, the deepest retrieved soil was used)."*

**As regards the results section – the main query is the categorical approach to representing the data in Figure 2 and Figure 3 and using the exponential and linear regression (respectively). The categorical approach can stand alone without the regressions, however if using regressions then the x axis should be represented as a continuous variable. Please consider revising this.**

In the revised manuscript we have removed all regressions and instead use the categorical approach to comparing isotope data for the thermal fractions (Figures 1 and 2, **L170** and **L193**, respectively).

**In terms of the two tables in the main body of the manuscript -is it possible to graphically represent some of this data? I think much like the graphical abstract, there is scope to conceptualize the data in diagrammatic form to increase the impact of the findings.**

The data from these two tables was presented in the original version of the manuscript in Figures 2 and 3, and in the revised manuscript Figures 1 (**L170**) and 2 (**L193**). In the revised manuscript we have improved our data visualisation to increase the impact and clarity of our findings.

**Is there data or any published work on the carbon accumulation rates at this site? How do the authors think the findings of this study would integrate with C org accumulation rate data.**

Yes, there are published carbon (C) accumulation rates for this site (Miller et al., 2023a; Smeaton et al., 2024). Compared to other Scottish and UK saltmarshes, Skinflats has relatively high C accumulation rates (see Miller et al., 2023; Smeaton et al., 2024). We have added detail on the implications of this to our discussion section.

**L335-356:** *"Compared to other UK saltmarshes, Skinflats has relatively high C accumulation rates (Miller et al., 2023; Smeaton et al., 2024). Depleted $^{14}C$ contents of the OC accumulating at*

the Skinflats saltmarsh (Houston et al., 2024b) imply that a proportion of the OC being buried may already have been aged at the time of deposition on the marsh surface, as the marsh formed in the 1930's (Miller et al., 2023). The combination of high carbon accumulation rates and depleted soil $^{14}C$ contents implies that the Skinflats saltmarsh accumulates a high proportion of old, most likely allochthonous OC. Some of the aged, allochthonous OC may have undergone significant microbial processing and degradation prior to its accumulation in the saltmarsh soil. As the OM is degraded, and the energetically favourable components are consumed, the resulting OM becomes increasingly thermally recalcitrant (Luk et al., 2021; Sanderman and Grandy, 2020; Soldatova et al., 2024). The accumulation of a high proportion of degraded OC on the Skinflats saltmarsh may therefore explain the lack of observed change in the isotopic composition of the soil OC pools with depth.

Not all old OC is degraded or thermally recalcitrant, and our results show that the Skinflats saltmarsh is also a store of old ($^{14}C$-depleted), thermally labile OC (Fig. 1). Old OC can be thermally labile if it 'ages' (is stored) in an environment with low decomposition rates, e.g., a peatland (Dean et al., 2023), prior to transport and accumulation into the saltmarsh. There are extensive peatlands in the Skinflats catchment, many of which are degrading (Lilly et al., 2012). Regardless of the age and degradation state of the OC deposited onto the marsh surface, as it gets buried it will undergo a degree of microbial processing and degradation in the saltmarsh soil (Luk et al., 2021), but that process is potentially less prevalent at Skinflats than saltmarshes accumulating younger, less degraded OC."

**As these samples are taken from the lower shore of the saltmarsh site can the authors speculate on how the results could vary with proximity to the terrestrial border and the greater proportion of organic carbon from autochthonous sources?**

We have added detail on this to the revised manuscript discussion.

**L455-466:** "*The samples used for this study were from the low marsh zone only, but it is likely that the thermal reactivity of the Skinflats saltmarsh soil C will vary spatially across the marsh, as the proportion of OC sources has been shown to be variable across saltmarshes (Middelburg et al., 1997). Given our findings that old ($^{14}C$-depleted) OC has greater thermal recalcitrance than young ($^{14}C$-enriched) OC (Fig. 1), we anticipate that higher marsh zones, which typically have greater proportions of autochthonous OC than lower marsh zones (Spohn et al., 2013), would contain a greater proportion of thermally labile OC. However, it is important to recognise that some of the young ($^{14}C$-enriched), autochthonous OC in saltmarsh soils can also be thermally recalcitrant. As well as marsh zonation, we expect that the proportion of OC sources (and associated mix of thermal reactivities) would also vary with proximity to marsh creeks which redistribute autochthonous and allochthonous C across the saltmarsh habitat (Middelburg et al., 1997; Reed et al., 1999).*"

**The abstract concludes with a sentence on how this study has relevance for saltmarsh management and therefore I expected a portion of the discussion to focus on this but I found this was just briefly mentioned at the end of the discussion and conclusion sections. I think greater discussion of the practical implications that the findings may have is necessary.**

We welcome the reviewer's encouragement to expand on the implications of our results and have added an '*implications*' section to the discussion in the revised manuscript.

**L414-472:** "*Our results show that aged (presumed allochthonous), thermally labile OC stored in saltmarsh soils remains vulnerable to loss to the atmosphere upon habitat drainage. Saltmarsh soils usually exist in low-oxygen, tidally-inundated conditions which slow decomposition of OC (Chapman et al., 2019), but many saltmarshes globally have been drained (and their soils*

[revised manuscript text omitted]

*In this paper we have determined that age ($^{14}$C-content) is related to the thermal recalcitrance of saltmarsh soil OC. We therefore speculate that sites accumulating younger OC would have more thermally labile soil OC than sites accumulating older OC, like Skinflats, with wider implications for the risks to these vulnerable stores of soil carbon from human disturbances."*

We have also adjusted our conclusion section to better convey the impact of our findings.

**L474-487:** "*This is the first study on saltmarsh soils to employ the ramped oxidation method. We show that old ($^{14}$C-depleted) carbon dominates the thermally recalcitrant OC pools. The thermally labile OC pools are also aged ($^{14}$C-depleted) compared to the contemporary atmosphere but are younger than the thermally recalcitrant OC pools. These results highlight the role of saltmarshes as mixed stores of both old, thermally recalcitrant OC, as well as younger, thermally labile OC.*

*We present the first comparison of the bioavailability ($CO_2$ evolved from incubation experiments; Houston et al., 2024)) and the thermal reactivity (RO) of saltmarsh soil OC. We show that aged, allochthonous $CO_2$ evolved from saltmarsh soils exposed to oxic conditions (Houston et al., 2024b) are from a predominantly thermally labile OC pool. As saltmarsh soils exist mostly in low oxygen, waterlogged conditions, management interventions to limit their exposure to elevated oxygen availability may protect and conserve these stores of thermally labile OC and provide a climate abatement service. Therefore, we recommend that thermally labile allochthonous OC stored in saltmarsh soils should be counted as additional in some carbon crediting projects and National GHG Inventories."*

Yours sincerely,

Alex Houston (on behalf of all authors).

---

## Referee Report (RR1)

*Review of Houston et al. "Old Carbon, New Insights: Thermal Reactivity and Bioavailability of Saltmarsh Soils" (Biogeosciences; https://doi.org/10.5194/egusphere-2024-3281)*

Synopsis

      This is a revised version of a previous manuscript focusing on Ramped Oxidation (RO) $^{14}$C activities (reported as "percent modern" or pMC) and $\delta^{13}$C values for organic carbon (OC) from a set of saltmarsh soil cores from the Skinflats saltmarsh in Scotland, UK. Based on my comments and those of a second, anonymous reviewer, the authors have made considerable changes to the revised version. I believe this revised version represents a significant improvement, but I still have several issues, particularly related to the treatment and presentation of the inverse model results. I highlight these in detail below---it is not clear to me that the authors fully comprehend what is being calculated and reported in these inversions. Only after implementing these further changes would I then support publication in *Biogeosciences*. I believe this will require one more round of review. Please do not hesitate to contact me regarding any questions on this review.

Sincerely,

Jordon Hemingway
jordon.hemingway@eaps.ethz.ch

(there are no line numbers given for the abstract, so I will just write my comments here and the authors can find the relevant lines).

- "…driven by the net contribution from the older fraction…": What does this mean? Are the authors saying that preserved OC in salt marshes is generally low in $^{14}C$ activity?

- "We also present the first evidence to supprt…": I'll admit that I'm not very well-versed in the MRV side of CDR, but it seems wild to me to claim that thermally labile OC that is currently preserved in salt marshes could count as *additional* $CO_2$ removal. This is carbon that is already sequestered. I understand that draining and disturbing these salt marshes would lead to *remineralization* of this OC, thus increasing $CO_2$ emissions, but doing nothing will not lead to any *additional* $CO_2$ being removed from the atmosphere. What am I missing here?

(here beings the line numbers)

L27: Hemingway et al. (2017) used an oxidizing carrier gas and should thus be cited along with Plante et al. and Stoner et al., not with Rosenheim et al.

L30: "$CO_2$ evolved at low temperatures is deemd to be from… pools with greater thermal lability than $CO_2$ evolved at higher temperatures". Yes, of course it is---this is the definition of thermal lability! I don't see what sentences like this are adding.

L34 (and throughout): change to "$^{14}C$ *activity*", as it is a radioactive isotope.

L57-59: "Crucially, the biological availability… depends on… thermal reactivity". This is not true. Biological availability may *correlate* with thermal reactivity, but it does not depend on it *per se*.

L108-109: "stream of high purity oxygen": I suppose I didn't realize in the first round of review that this is indeed a pure $O_2$ stream (I now dug into the Gartnett et al. 2023 paper). Given this, do the authors think this difference in carrier gas will impact thermogram shape relative to other systems? Have they compared a reference material using their setup vs. using the setup at NOSAMS, ETH, etc.? It would be really nice to see the inter-laboratory reproducibility of this instrument (not just internal reproducibility, which looks quite nice in Gartnett et al.). For reference, most other systems use $O_2$ in He. As a starting point for this comparison, the authors could look into Bolandini et al. (2025) *Radiocarbon* (https://doi.org/10.1017/RDC.2025.6), who investigated the impact of $O_2$ flow rate on thermogram shape for the ETH instrument.

L151-153: Fair point to the authors in their response that this is indeed how most labs used to do a $^{13}C$ correction prior to AMS instruments including a $^{13}C$ cup---I did not realize this is still how things are doing at SUERC. Still, I note that the authors mis-cited McIntyre et al. (2017) in their response, who indeed used the internal $^{13}C$ correction of the MiCaDaS system, as is common practice at ETH (that paper instead focuses on the in-line EA-IRMS-AMS for single-analysis %OC, $\delta^{13}C$, and $F^{14}C$ analysis).

L159-162 (and thorughout): Please use $\mu_E$, $\sigma_E$, and $p(0, E)$ nomenclature.

L162-166 (and Table A2): I don't understand this---one can simply tell the software to perform a blank correction or not using the "blank_corr" flag (see documentation). How do the "$^{13}C$

values generated [vary] significantly from our IRMS measured values"? If the software does not perform a blank correction, then the $\delta^{13}C$ values used by the software are simply identical to the ones inputted by the user---there is nothing to be "modelled" here. I don't understand how the numbers in Table A2 were generated.

Fig. 1-2/Table 1-4: (I'm not sure exactly where to put this comment, so I will put it at the first place that I think is relevant, which is Fig. 1.) Here, the authors need to report much more information related to the inversion before these results can be interpreted. For example:

- what regularization values, $\lambda$, were used for each sample?
- How do the resulting $p(0, E)$ distributions for each sample look? The authors show the thermograms, but never show $p(0, E)$ distributions. It is difficult to judge results without seeing the distributions themselves. This is particularly the case since some of the samples appear to not reach baseline at high temperature (e.g., T3 5.5, T2 15.5). This is important as it is known that the inversion is sensitive to boundary effects, so baseline must by reached or forced (see documentation).
- It would be incredibly useful for the subsequent discussion to know what fraction of each sample is contained within each thermal window. For example, is the 150-325 °C fraction 10% of total C? 20%? This sould be added, e.g., to Table 1 or 2.
- I am again missing the a comparison of mass-weighted RO results vs. bulk measured results. That is, if you simply sum the $F^{14}C$ or $\delta^{13}C$ values for each thermal window weighted by the fraction of total carbon within that thermal window, do you recreate the measured bulk values within uncertainty? This is again an important "sanity check" and can easily be added, e.g., to Tables 1-2 (or Table A3).
- In Table 3, what does the $p(0, E)$ column mean? $p(0, E)$ is a probability density function---that is, a distribution whose integral is equal to unity. It is not a single scalar number. I don't know what 0.02, 0.02, 0.01, etc. refer to. Is this the maximum value in the $p(0, E)$ distribution? But this is arbitrary and depends on the size of the discretized energy step, $\Delta E$…
- Table 4 (and throughout): please update the nomenclature so that it is clear to the reader when the authors are referring to $\mu_E$, $\sigma_E$, and $p(0, E)$ of the entire sample vs. for a given thermal window; following previous studies, I recommend $\mu_E$, $\sigma_E$, and $p(0, E)$ when referring to the bulk sample and $\mu_{f,E}$, $\sigma_{f,E}$, and $\Pi_f(E)$ when referring to a given thermal window, $f$.

L222-223: "…no significant changes in $\mu_E$, $\sigma_E$, nor activation energy distribtuion ($p(0, E)$)". But $p(0, E)$ is a distribution, not a scalar value, so how can it be compared across samples in the same way as the other metrics?

L232: "…$\mu_{f,E}$ [here using my recommended nomenclature]… increased sequentially…we therefore infer that the thermal recalcitrance of RO fractions is greater at higher temperatures…". As for my comment on L30, of course it is---this is the definition of thermal recalcitrance! Higher temperature thermal fractions will have a higher $\mu_{f,E}$ value by definition---there is nothing to infer!

L252: What is meant by "…wider $\mu_E$ ranges…compared to the first three RO fractions"? $\mu_{f,E}$ [here using my recommended nomenclature] is a mean value so it cannot have a "range". Do the authors mean that the *difference* in $\mu_{f,E}$ is greater between the highest two thermal windows

than between the lowest three? But if so, then this is simply a function of the chosen temperature windows and doesn't say anything inherent about the OC being combusted.

L253-254: "…may have been caused by non-first order decomposition of carbonates". How would this cause "wider $\mu_E$ ranges… and increased bond strength diversity"? Non-first order behavior implies that the resulting thermogram (and thus $p(0, E)$) shape depends on the mass of sample loaded into the instrument (c.f., Fig. 4d of Hemingway et al. (2017) *Biogeosciences*).

L255: change "*rampedpyrox* model" to "distributed actvation energy model"; *rampedpyrox* is simply the name of the python package.

L298-299: "…although the thermal reactivity of OC decreases with $^{14}$C content…" The cause-and-effect should be flipped here: $^{14}$C activity decreases with decreasing thermal reactivity (i.e., thermal reactivity is the independent variable).

L344-347: Here the logic seems to be: (i) low-$E$ components are consumed prior to deposition in the Skinflats; this leads to (ii) thermally recalcitrant material being deposited and thus (iii) no change with depth in the salt marsh (i.e., due to no further remineralization). But this somewhat contradicts the thermograms shown in Fig. 1 (and presumably the corresponding $p(0, E)$ distributions, if they were shown), which shows a fair amount of carbon in the ~200-400 °C range. This is quite thermally labile. In fact, one *does* observe a decrease in the peak height of the ~250 °C peak with depth (relative to the ~450 °C peak). This instead points to a continued remineralization of this thermally labile material with burial depth. This is one instance where it would be very useful to know what fraction of total carbon is contained within each thermal window, as this could then be easily quantified. An alternative approoch is to use the fraction of total carbon contained in "low-$E$", "middle-$E$", and "high-$E$" bins, as was done for example in Hemingway et al. (2018) *Science* (https://doi.org/10.1126/science.aao6463).

L360-361: "…more energy is required …to decompose older…carbon than younger…carbon". Careful with statements like this; thermal activation energy is merely an analytical tool to separate carbon; there is no requirement that older carbon necessarily has a higher thermal recalcitrance.

L365-266: Schmidt et al. (2011) do not mention thermal reactivity. Also, biological turnover time does not *depend* on thermal reactivity *per se*; the latter is merely an analytical tool.

L380-382: This statement is well-known in the RPO literature (see, e.g., some of the initial papers from the Rosenheim group that focused on using RPO as a means of dating sediments).

L389-390: I don't think you can say that, "…the biologically evolved $CO_2$…was therefore not from a thermally labile OC pool." It very well could have been from a labile pool if said pool was composed of several compounds of different $^{14}$C ages, as is likely.

L399-401: I don't know what this sentence is trying to say. Reword.

L432-433: This relates to my comment in the abstract, but can this *really* be considered as additional C storage for MRV? This carbon is already naturally sequestered…

L485-487: Same as previous comment.

---

## Author Response (AR2)

Dear Editor,

Thank you for your feedback and decision, inviting us to resubmit with minor revisions on our manuscript, "*Old Carbon, New Insights: Thermal Reactivity and Bioavailability of Saltmarsh Soils*", submitted on 21/10/2024; revised and resubmitted on 07/05/2025; further revised and resubmitted on 14/07/2025.

We are now pleased to provide an updated and further revised manuscript for your consideration for publication in EGU *Biogeosciences*.

Below, please find our detailed responses to the second round of reviewer 1 comments. We have attached two copies of the revised manuscript, one with tracked changes and one cleaned, final version.

We believe that this revised manuscript is improved and that we have fully addressed all comments.

We now look forward to receiving your decision on its publication in *EGU Biogeosciences*.

Yours sincerely,

Alex Houston, on behalf of all co-authors

**Review of Houston et al. "Old Carbon, New Insights: Thermal Reactivity and Bioavailability of Saltmarsh Soils" (Biogeosciences; https://doi.org/10.5194/egusphere-2024-3281)**

**Synopsis**

This is a revised version of a previous manuscript focusing on Ramped Oxidation (RO) 14C activities (reported as "percent modern" or pMC) and d13C values for organic carbon (OC) from a set of saltmarsh soil cores from the Skinflats saltmarsh in Scotland, UK. Based on my comments and those of a second, anonymous reviewer, the authors have made considerable changes to the revised version. I believe this revised version represents a significant

improvement, but I still have several issues, particularly related to the treatment and presentation of the inverse model results. I highlight these in detail below---it is not clear to me that the authors fully comprehend what is being calculated and reported in these inversions. Only after implementing these further changes would I then support publication in Biogeosciences. I believe this will require one more round of review. Please do not hesitate to contact me regarding any questions on this review.

Sincerely,

Jordon Hemingway

jordon.hemingway@eaps.ethz.ch

We thank the reviewer for their second review of this manuscript, which we agree is much improved from the original submission. We have added detailed responses to each individual query below. We believe that we have fully addressed each query and look forward to the editor's decision on its publication.

(there are no line numbers given for the abstract, so I will just write my comments here and the authors can find the relevant lines).

• "...driven by the net contribution from the older fraction...": What does this mean? Are the authors saying that preserved OC in salt marshes is generally low in 14C activity?

Saltmarshes accumulate OC from different autochthonous (in-situ production) and allochthonous (externally derived) sources, which can be differently aged. Hence, a saltmarsh soil sample can contain both 'young' and 'old' components. This turnover of differently aged OC in saltmarsh soils is a key part of this work.

For example, the accumulation of differently aged OC is already mentioned in the previous sentence of the abstract:

"... *accumulate organic carbon from both modern and aged sources through in-situ biological production and the capture of ex-situ sources which are deposited during tidal inundation.*"

• "We also present the first evidence to supprt...": I'll admit that I'm not very well-versed in the MRV side of CDR, but it seems wild to me to claim that thermally labile OC that

is currently preserved in salt marshes could count as additional CO2 removal. This is carbon that is already sequestered. I understand that draining and disturbing these salt marshes would lead to remineralization of this OC, thus increasing CO2 emissions, but doing nothing will not lead to any additional CO2 being removed from the atmosphere.

What am I missing here?

The reviewer is correct that thermally labile OC preserved in a saltmarsh is not 'additional' in a business-as-usual scenario. However, management interventions which reduce the emission of stored OC to the atmosphere (e.g., protection from a degradation pressure) can in some cases be counted as delivering reduced emissions. A foundation of this manuscript is the evidence from Houston et al. (2024) that the Skinflats saltmarsh was respiring $CO_2$ from $^{14}C$ depleted OC pools in a simulated drainage degradation scenario. In this manuscript, we show that the respired $^{14}C$-$CO_2$ content from the above study was closest (in most cases) to the $^{14}C$ content of the most thermally labile OC pool. Therefore, we propose that the thermally labile OC pool in saltmarsh soils which are protected against a drainage degradation pressure for the purpose of generating carbon credits or contributing to national Greenhouse Gas Inventories, could be counted as delivering emissions reduction.

We direct interested readers to the following helpful resource:

Griscom, Bronson W., et al. "Natural climate solutions." *Proceedings of the National Academy of Sciences* 114.44 (2017): 11645-11650.

The above argument is already covered in the abstract:

"*Management interventions (e.g. rewetting by tidal inundation) to limit the exposure of saltmarsh soils to elevated oxygen availability may help to protect and conserve these stores of thermally labile organic carbon and hence limit $CO_2$ emissions.*"

We have added further clarification to the abstract:

"*We also present evidence to support the inclusion of thermally labile allochthonous OC stored in saltmarsh soils in additionality assessments for projects which aim to prevent the drainage of saltmarshes...*"

(here beings the line numbers)

L27: Hemingway et al. (2017) used an oxidizing carrier gas and should thus be cited along with Plante et al. and Stoner et al., not with Rosenheim et al.

We thank the reviewer for this clarification and have updated the text.

L26: "*(e.g., Hemingway et al., 2017b; Plante et al., 2011; Stoner et al., 2023), or other gases, typically Helium (e.g., Rosenheim et al., 2008).*"

L30: "CO2 evolved at low temperatures is deemd to be from... pools with greater thermal lability than CO2 evolved at higher temperatures". Yes, of course it is---this is the definition of thermal lability! I don't see what sentences like this are adding.

We feel that it is important to define this for readers, who may not be experts in the field. We have changed 'deemed' to 'derived' to clarify that this happens.

L30: "$CO_2$ evolved at low temperatures is derived from soil OC pools with a greater thermal lability than $CO_2$ evolved at higher temperatures"

L34 (and throughout): change to "14C activity", as it is a radioactive isotope.

"$^{14}C$ content" is standard in the reporting of radiocarbon as fraction or percentage modern, and we have retained this throughout the manuscript. We think it would be fine to use activity, but that this would add unnecessary confusion for readers when $^{14}C$ content is already being used as consistent terminology.

L57-59: "Crucially, the biological availability... depends on... thermal reactivity". This is not true. Biological availability may correlate with thermal reactivity, but it does not depend on it per se.

This is fair, we have changed 'depends on' to 'related to' throughout the revised manuscript when referring a relationship between thermal reactivity and biological availability.

L108-109: "stream of high purity oxygen": I suppose I didn't realize in the first round of review that this is indeed a pure O2 stream (I now dug into the Gartnett et al. 2023 paper). Given this, do the authors think this difference in carrier gas will impact thermogram shape relative to other systems? Have they compared a reference material using their setup vs. using the setup at NOSAMS, ETH, etc.? It would be really nice to see the inter-laboratory reproducibility of this instrument (not just internal reproducibility, which looks quite nice in Gartnett et al.). For reference, most other systems use O2 in He. As a starting point for this comparison, the authors could look into Bolandini et al. (2025) Radiocarbon (https://doi.org/10.1017/RDC.2025.6), who investigated the impact of O2 flow rate on thermogram shape for the ETH instrument.

*This is an interesting suggestion but is not within the scope of this study and not directly relevant to our findings.*

L151-153: Fair point to the authors in their response that this is indeed how most labs used to do a 13C correction prior to AMS instruments including a 13C cup---I did not realize this is still how things are doing at SUERC. Still, I note that the authors mis-cited McIntyre et al. (2017) in their response, who indeed used the internal 13C correction of the MiCaDaS system, as is common practice at ETH (that paper instead focuses on the in-line EA-IRMS-AMS for singleanalysis %OC, d13C, and F14C analysis).

*We do not agree that this is mis-cited. We were making the point that some labs use IRMS $^{13C}$ values for normalising $^{14}$C results. Indeed, McIntyre et al. (2017) state in their Introduction "*In some laboratories, offline IRMS sample δ$^{13}$C is used for retroactive $^{14}$C correction calculations*". P.894*

L159-162 (and thoroughout): Please use µE, sE, and p(0, E) nomenclature.

*We have implemented this change in nomenclature throughout.*

L162-166 (and Table A2): I don't understand this---one can simply tell the software to perform a blank correction or not using the "blank_corr" flag (see documentation). How do the "13C values generated [vary] significantly from our IRMS measured values"? If the software does not perform a blank correction, then the d13C values used by the software are simply identical to the ones inputted by the user---there is nothing to be "modelled" here. I don't understand how the numbers in Table A2 were generated.

*We are grateful to the reviewer for informing us on how to disable the blank correction in the* rampedpyrox *software. We have therefore removed the sentence and Table A2 as they are not required.*

Fig. 1-2/Table 1-4: (I'm not sure exactly where to put this comment, so I will put it at the first place that I think is relevant, which is Fig. 1.) Here, the authors need to report much more information related to the inversion before these results can be interpreted. For example:

• what regularization values, l, were used for each sample?

*We used the best-fit values which were generated by the model based on our data, following the model documentation. We have added these values to the supplementary information (rp_outputs -> sample -> Figure_2).*

• How do the resulting p(0, E) distributions for each sample look? The authors show the thermograms, but never show p(0, E) distributions. It is difficult to judge results without seeing the distributions themselves. This is particularly the case since some of the samples appear to not reach baseline at high temperature (e.g., T3 5.5, T2 15.5). This is important as it is known that the inversion is sensitive to boundary effects, so baseline must by reached or forced (see documentation).

*We have added these distributions to the supplementary information (rp_outputs -> sample -> Figure_3). Per model documentation, we forced the baseline to be reached. This does not change our findings.*

• It would be incredibly useful for the subsequent discussion to know what fraction of each sample is contained within each thermal window. For example, is the 150-325 °C fraction 10% of total C? 20%? This sould be added, e.g., to Table 1 or 2.

*We have added this table to the supplementary information (Table A3), for both the total C released for each temperature fraction and the % of each sample which was evolved for each temperature fraction. There were no significant changes in the proportion of C released from each fraction with burial depth and we have added a sentence on this to the discussion to clarify:*

*L338: "This interpretation is supported by the lack of change in both the amount and the proportion of $CO_2$ evolved from each change temperature fraction with depth (ANOVAs, p > 0.05. Table A3)."*

• I am again missing the a comparison of mass-weighted RO results vs. bulk measured results. That is, if you simply sum the F14C or d13C values for each thermal window weighted by the fraction of total carbon within that thermal window, do you recreate the measured bulk values within uncertainty? This is again an important "sanity check" and can easily be added, e.g., to Tables 1-2 (or Table A3).

*We cannot do this 'sanity check' for these samples because, as stated, the bulk soils were acid-treated, whilst the RO samples were not (as discussed in our previous response letter), so we would not be comparing like-for-like.*

Although we can't do this check for these samples due to the different pretreatments, previous work using this analytical set-up have done this for other samples and shown that the combined ROx fractions do equal the bulk isotope values (Garnett et al 2023).

Garnett, M. H., et al. "A new ramped oxidation-14C analysis facility at the NEIF Radiocarbon Laboratory, East Kilbride, UK." *Radiocarbon* 65.5 (2023): 1213-1229.

We have included above explanation to the revised manuscript; see:

L186: "*The RO samples were not pre-treated with acid, but the samples for bulk soil-$^{14}$C were (Houston et al., 2024b), so we cannot verify that the weighted RO-$^{14}$C contents amassed to the bulk soil $^{14}$C content. However, previous work using this analytical set-up have done this for other samples and shown that the combined RO fractions do equal the bulk isotope values (Garnett et al., 2023).*"

• In Table 3, what does the p(0, E) column mean? p(0, E) iss a probability density function-

--that is, a distribution whose integral is equal to unity. It is not a single scalar number.

I don't know what 0.02, 0.02, 0.01, etc. refer to. Is this the maximum value in the p(0,

E) distribution? But this is arbitrary and depends on the size of the discretized energy

step, ΔE...

p(O, E) refers to the maximum value of the probability density distribution. In the revised text, we have removed the p(O, E) column from Table 3 and from the text; we acknowledge that it was not a helpful addition.

• Table 4 (and throughout): please update the nomenclature so that it is clear to the reader

when the authors are referring to µE, sE, and p(0, E) of the entire sample vs. for a given

thermal window; following previous studies, I recommend µE, sE, and p(0, E) when

referring to the bulk sample and µf,E, s f,E, and Pf (E) when referring to a given thermal

window, f.

We have now made these requested nomenclature changes throughout the revised manuscript.

L222-223: "...no significant changes in µE, sE, nor activation energy distribtuion (p(0, E))".

But p(0, E) is a distribution, not a scalar value, so how can it be compared across samples in

the same way as the other metrics?

We have removed the discussion of p(O, E) trends from the revised manuscript, without altering the interpretation of the overall results. We thank the reviewer for pointing out this mistake.

L232: "…µf,E [here using my recommended nomenclature]… increased sequentially…we therefore infer that the thermal recalcitrance of RO fractions is greater at higher temperatures…". As for my comment on L30, of course it is---this is the definition of thermal recalcitrance! Higher temperature thermal fractions will have a higher µf,E value by definition---there is nothing to infer!

We have removed the quoted text from the revised manuscript as we agree that thermal recalcitrance is greater at higher temperatures. This is a point we make in the introduction to aid the reader and it does not need to be repeated here.

L252: What is meant by "…wider µE ranges…compared to the first three RO fractions"? µf,E [here using my recommended nomenclature] is a mean value so it cannot have a "range". Do the authors mean that the difference in µf,E is greater between the highest two thermal windows than between the lowest three? But if so, then this is simply a function of the chosen temperature windows and doesn't say anything inherent about the OC being combusted.

This sentence has been removed from the revised manuscript because the model results do not show any significant trends and are redundant to further discussion.

L253-254: "…may have been caused by non-first order decomposition of carbonates". How would this cause "wider µE ranges… and increased bond strength diversity"? Non-first order behavior implies that the resulting thermogram (and thus p(0, E)) shape depends on the mass of sample loaded into the instrument (c.f., Fig. 4d of Hemingway et al. (2017) Biogeosciences).

Because of the equivocal nature of these results and the fact that they are not necessary to support our overall findings, we have removed this discussion point as being redundant.

L255: change "rampedpyrox model" to "distributed activation energy model"; rampedpyrox is simply the name of the python package.

We have made this change throughout.

L298-299: "…although the thermal reactivity of OC decreases with 14C content…" The causeand-effect should be flipped here: 14C activity decreases with decreasing thermal reactivity (i.e., thermal reactivity is the independent variable).

We have made this change.

L289: "*although $^{14}C$ content decreases with decreasing thermal reactivity*"

L344-347: Here the logic seems to be: (i) low-E components are consumed prior to deposition in the Skinflats; this leads to (ii) thermally recalcitrant material being deposited and thus (iii) no change with depth in the salt marsh (i.e., due to no further remineralization). But this somewhat contradicts the thermograms shown in Fig. 1 (and presumably the corresponding p(0, E) distributions, if they were shown), which shows a fair amount of carbon in the ~200-400 °C range. This is quite thermally labile. In fact, one does observe a decrease in the peak height of the ~250 °C peak with depth (relative to the ~450 °C peak). This instead points to a continued remineralization of this thermally labile material with burial depth. This is one instance where it would be very useful to know what fraction of total carbon is contained within each thermal window, as this could then be easily quantified. An alternative appraoch is to use the fraction of total carbon contained in "low-E", "middle-E", and "high-E" bins, as was done for example in Hemingway et al. (2018) Science (https://doi.org/10.1126/science.aao6463).

We agree that there is clearly some thermally labile C in the samples, this is already discussed in the next paragraph.

L341: "*Not all old OC is degraded or thermally recalcitrant, and our results show that the Skinflats saltmarsh is also a store of old ($^{14}C$-depleted), thermally labile OC (Fig. 1).*"

Whilst we agree that visually it looks like there may be trends in the proportion of C evolved from each temperature fraction with depth, and we had discussed this in our original submission, there were no significant trends with depth in either the amount or the proportion of C evolved from each temperature fraction. We have added this data to the supplementary information (Table A3) and added a sentence to the discussion.

L338: "*This interpretation is supported by the lack of change in both the amount and the proportion of $CO_2$ evolved from each change temperature fraction with depth (ANOVAs, $p > 0.05$. Table A3).*"

L360-361: "…more energy is required …to decompose older…carbon than younger…carbon".

Careful with statements like this; thermal activation energy is merely an analytical tool to

separate carbon; there is no requirement that older carbon necessarily has a higher thermal recalcitrance.

We agree with the reviewer for the need to be careful here, but feel it is important to state that our findings are consistent with much previous research on different soil/sediment systems. We have altered the text to state '*in most cases*' to make it clear that it is not a requirement that older carbon has a greater thermal recalcitrance than younger carbon.

L353: "*which have found that in most cases, more energy is required (higher temperature/μE) to decompose older ($^{14}$C-depleted), degraded/microbially derived ($^{13}$C-enriched) C than younger ($^{14}$C-enriched), less processed ($^{13}$C-depleted) C*"

L365-266: Schmidt et al. (2011) do not mention thermal reactivity. Also, biological turnover time does not depend on thermal reactivity per se; the latter is merely an analytical tool.

We have changed 'depends on' to 'related to' and removed the Schmidt et al. (2011) reference. No reference is required as we have stated this fact previously in the manuscript.

L359: "*As the biological turnover time of OC is related to the prevailing environmental conditions as well as thermal reactivity*"

L380-382: This statement is well-known in the RPO literature (see, e.g., some of the initial papers from the Rosenheim group that focused on using RPO as a means of dating sediments).

We are aware, but we still need to report it as it occurred for our samples. We have added a sentence stating that this has been found in previous research.

L376: "*Similar findings of mixing within thermal fractions has been reported in previous RPO work (e.g., Rosengard et al., 2025, Rosenheim et al., 2008).*"

L389-390: I don't think you can say that, "...the biologically evolved CO2...was therefore not from a thermally labile OC pool." It very well could have been from a labile pool if said pool was composed of several compounds of different 14C ages, as is likely.

We have altered the wording to make it clear that the pool was potentially not from a thermally labile pool, although it is possible that it was as all pools are mixtures.

L385: "*... potentially derived from less thermally labile OC pools than the other samples, although it is possible that the thermally labile pools were composed of multiple OC sources with different $^{14}$C contents.*"

L399-401: I don't know what this sentence is trying to say. Reword.

Reworded.

L396: "*Degradation of some of the thermally labile OM components during burial may reduce the range of differently aged OC sources within the most thermally labile RO fraction for the deeper samples in this study.*"

L432-433: This relates to my comment in the abstract, but can this really be considered as

additional C storage for MRV? This carbon is already naturally sequestered...

See our reply to the comment on the abstract. This is covered in the previous paragraph.

L420: "*Protecting saltmarshes from degradation following drainage is listed as an eligible activity for generating carbon credits for blue carbon ecosystem (BCE) projects (VERRA, 2023) and there is significant potential for climate mitigation by avoided emissions from protecting vulnerable stocks of soil OC in BCEs (Goldstein et al., 2020; Griscom et al., 2017; Kwan et al., 2025; Sasmito et al., 2025). Similarly, the re-creation of saltmarsh habitat through managed realignment (rewetting by tidal inundation) of historic saltmarsh habitats which were previously reclaimed for land use purposes (e.g., agriculture) could reduce (and possible reverse) the emissions of aged OC to the atmosphere, both locally to Skinflats, and globally.*"

L485-487: Same as previous comment

See above.